# SUBLINEAR ITERATIONS CAN SUFFICE EVEN FOR DDPMS

## ABSTRACT

SDE-based methods such as denoising diffusion probabilistic models (DDPMs) have shown remarkable success in real-world sample generation tasks. Prior analyses of DDPMs have been focused on the exponential Euler discretization, showing guarantees that generally depend at least linearly on the dimension or initial Fisher information. Inspired by works in log-concave sampling (Shen & Lee, 2019), we analyze an integrator – the denoising diffusion randomized midpoint method (DDRaM) – that leverages an additional randomized midpoint to better approximate the SDE. Using a recently-developed analytic framework called the "shifted composition rule", we show that this algorithm enjoys favorable discretization properties under appropriate smoothness assumptions, with sublinear $\widetilde{O}(\sqrt{d})$ score evaluations needed to ensure convergence. This is the first sublinear complexity bound for pure DDPM sampling — prior works which obtained such bounds worked instead with ODE-based sampling and had to make modifications to the sampler which deviate from how they are used in practice. We also provide experimental validation of the advantages of our method, showing that it performs well in practice with pre-trained image synthesis models.

## 1 INTRODUCTION

With the emergence of diffusion models (Sohl-Dickstein et al., 2015; Song & Ermon, 2019; Ho et al., 2020; Song et al., 2021b) as the leading paradigm for generative modeling in image (Rombach et al., 2022), video (Ho et al., 2022; Blattmann et al., 2023), and molecular generation (Geffner et al., 2025a;b), a flurry of recent work has sought to place these models on rigorous footing using mathematical insights from high-dimensional statistics and numerical analysis. An early finding in this line of work was that, given sufficiently accurate score estimation, diffusion models can sample from essentially any probability distribution in $d$ dimensions in $O(d)$ *iterations* (Chen et al., 2023c; Lee et al., 2023; Benton et al., 2024; Conforti et al., 2025a).

Subsequently, there has been sustained interest in quantitatively tightening this bound. A number of works (Chen et al., 2023b; Li et al., 2024b; Huang et al., 2025; Gupta et al., 2025; Jiao & Li, 2025; Li & Jiao, 2025) have proven that for *ODE-based* diffusion samplers, i.e., DDIMs (Song et al., 2021a), the lack of stochasticity enables the design and analysis of algorithms that only require a number of iterations that is *sublinear* in $d$. Other works have tried circumventing $O(d)$ complexity by instead bounding the *parallel* complexity of diffusion-based sampling (Chen et al., 2024; Gupta et al., 2025; Zhou & Sugiyama, 2025), or by showing that diffusion models can adapt to the *intrinsic dimension* of the distribution (Li & Yan, 2024; Boffi et al., 2025; Liang et al., 2025; Tang & Yan, 2025), offering speedups orthogonal to the original question of tightening the dimension dependence.

For this guiding question, however, remarkably the best known guarantee for *SDE-based* diffusion samplers, i.e., DDPMs (Ho et al., 2020), has remained $O(d)$. In this work, we ask:

> *Can SDE-based diffusion sampling provably achieve sublinear complexity?*

In practice, SDE-based sampling confers a number of advantages that make this question particularly salient. In image generation, although DDIMs outperform DDPMs in the few-step regime, the performance for the former quickly saturates while the performance for the latter continues to improve as the number of steps increases; see, e.g., Karras et al. (2022, Figure 4) and Song et al. (2021b);

Cao et al. (2023); Gonzalez et al. (2023); Nie et al. (2024); Deveney et al. (2025). This observation has been borne out across a range of model scales: even for large-scale latent diffusions, properly tuned SDE-based samplers often obtain higher performance than their deterministic counterparts (Ma et al., 2024). Stochasticity of the sampling steps also plays a crucial rule in leading protein diffusion models (Abramson et al., 2024; Geffner et al., 2025b) as a way to heuristically trade off between diversity and designability. In stochastic optimal control-based approaches to steering diffusion models (Domingo-Enrich et al., 2025), during fine-tuning it is necessary to work with an SDE-driven base generative process, and the complexity of sampling enters not just at inference time, but during the training of the control policy. Likewise, when using stochastic optimal control to transport a point mass to some target measure (Havens et al., 2025), it is trivially necessary to use stochastic dynamics to generate entropy.

So what would it take to break the $O(d)$ barrier? Intuition from the log-concave sampling literature suggests that doing so requires a more refined discretization scheme. One of the most powerful such schemes emerging from that line of work is the *randomized midpoint method* (Shen & Lee, 2019), which forms the backbone of state-of-the-art bounds for log-concave sampling (Altschuler & Chewi, 2024b; Altschuler et al., 2025). This method has also been used in several recent works on ODE-based diffusion sampling (Gupta et al., 2025; Jiao & Li, 2025; Li & Jiao, 2025). To reap the benefits of randomized discretization however, all of them crucially rely on the deterministic nature of the sampling dynamics, combined with periodic injections of noise that are convenient for establishing provable guarantees but which deviate significantly from how diffusion models are implemented in practice. Indeed, it was explicitly listed as an unresolved challenge in the conclusion of Jiao & Li (2025) to extend these analyses to pure DDPMs, and as we discuss in §4, this runs into a surprising range of new obstacles.

## 1.1 CONTRIBUTIONS

In this work, we overcome these obstacles and answer our guiding question in the affirmative. We craft a new analysis framework for DDPMs that successfully interfaces with the randomized midpoint method, allowing us to break the $O(d)$ barrier for SDE-based diffusion sampling. We first informally state our main guarantee:

**Theorem 1** (Informal, see Theorem 3)**.** *Let $\varepsilon > 0$, and let $\pi$ be a data distribution over $\mathbb{R}^d$ with bounded second moment. Suppose we have estimates $(\mathsf{s}_t)$ for its scores $(\nabla \log \pi_t)$ along the Ornstein–Uhlenbeck process that are $\widetilde{O}(\varepsilon)$-accurate in $L^2(\pi_t)$ and $L_t$-Lipschitz for $L_t \lesssim (1 - e^{-2t})^{-1}$. Then, there is a discretization of DDPM that samples from a distribution $\hat{\pi}$ that is $\varepsilon^2$-close in $\mathsf{KL}$ divergence to a distribution $\pi^{\mathsf{approx}}$ that is $\varepsilon$-close in $W_2$ to $\pi$, with no more than $\widetilde{O}(\sqrt{d}/\varepsilon)$ sampling steps.*

There are two main innovations over prior work. First, state-of-the-art guarantees for DDPMs (Benton et al., 2024; Conforti et al., 2025a; Li & Yan, 2025) required $O(d)$ sampling steps. Second, state-of-the-art guarantees for DD*I*Ms that achieved sublinear complexity had to fundamentally modify the sampling algorithm (see §1.2), whereas we simply work with the standard DDPM reverse process used in practice, suitably discretized.

Our specific choice of discretization, the randomized midpoint method, has been employed in prior work on DDIM sampling (Shen & Lee, 2019; Gupta et al., 2025; Jiao & Li, 2025; Li & Jiao, 2025), but we provide the first analysis for DDPMs. Traditionally, the advantages of this choice of discretization are clear at the level of coupling-based arguments that bound the $W_2$ distance between the true process and the sampler, but for general, non-log-concave distributions, such arguments cannot be run for too long without incurring exponential blowups. Existing analyses in the diffusion setting sidestep this by artificially injecting noise into the dynamics, allowing one to "restart" the coupling. Unfortunately, without this trick, prior methods for analyzing DDPMs – which are rooted in TV / KL-based analysis – seem to be fundamentally incompatible with randomized midpoint. To overcome this, we build upon the *shifted composition method* (Altschuler & Chewi, 2024a), a powerful new technique from the log-concave sampling literature that combines the advantages of coupling-based $W_2$ analysis with those of information-theoretic TV / KL analysis. We defer a more comprehensive overview of our techniques to §4.

**On the smoothness assumption.** The main caveat relative to the prior $O(d)$ guarantees for DDPMs is that we make a smoothness assumption. However, this assumption is weaker than what is made in almost all previous papers on DDIMs that achieve sublinear complexity (Chen et al., 2023b; Gupta

et al., 2025; Li & Jiao, 2025). Those works additionally required smoothness of the true scores, and furthermore the assumed bound was independent of noise scale $t$, whereas our bound on $L_t$ becomes increasingly weaker as $t \to 0$. The one exception is the recent result of Jiao & Li (2025) for DDIMs; see §1.2 for discussion.

In the absence of any smoothness assumptions, it has remained a central open question in this literature how to obtain sublinear complexity bounds with any score-based algorithm, even an ODE-based one. This is well out of scope of this work, the focus of which is instead on bringing our theoretical understanding of DDPMs closer to what is known for DDIMs.

### 1.2 COMPARISON TO PRIOR WORK

Below we describe relevant prior work in the theoretical study of diffusion models.

**Discretization analyses for DDPMs.** Early work on diffusion model theory focused on convergence guarantees for DDPMs (Block et al., 2020; De Bortoli, 2022; Lee et al., 2022; Liu et al., 2022), which culminated in the finding by Chen et al. (2023c); Lee et al. (2023) that they can sample from essentially arbitrary distributions in polynomial time given $L^2$-accurate score estimates. This was subsequently refined by Chen et al. (2023a) and finally by Benton et al. (2024); Conforti et al. (2025a) to show convergence in $O(d/\varepsilon^2)$ iterations to a distribution that is $\varepsilon^2$-close in KL to a slight noising of the data distribution. By Pinsker's inequality, this implies $\varepsilon$-closeness in TV, which Li & Yan (2025) later showed could be obtained using only $O(d/\varepsilon)$ iterations. With the exception of this last work, which exploited a subtle recursive bound on the TV error, all prior works giving convergence guarantees for general distributions relied on Girsanov's theorem.

There have also been a number of works on showing that DDPMs can adapt to low-dimensional structure in the data (see, e.g., Huang et al. (2024); Li & Yan (2024); Potaptchik et al. (2024); Boffi et al. (2025); Liang et al. (2025) and the references therein). These results show that $d$ in the above rates can effectively be replaced with some measure of the *intrinsic dimension* $k$ of the distribution; while this is technically "sublinear" in the dimension if $k = o(d)$, our sublinear complexity holds even if $k = \Theta(d)$. We leave as an interesting open question how to get $o(k)$ rates using DDPMs. Finally, we remark that there have been various works seeking to modify DDPMs to achieve accelerated rates as a function of $\varepsilon$ (see, e.g., Li & Cai, 2024; Li et al., 2024a; Wu et al., 2024).

**Discretization analyses for DDIMs.** As mentioned above, all known diffusion-based sampling guarantees achieving sublinear complexity are based on DDIM sampling. Chen et al. (2023b) obtained the first sublinear complexity bound of $O(L^2\sqrt{d}/\varepsilon)$ for ODE-based samplers under the assumption that the true scores and the score estimates are $L$-Lipschitz. Their algorithm follows the probability flow ODE but injects randomness by running an *underdamped Langevin corrector* at the end of every time window of length $O(1/L)$. We still refer to such samplers as ODE-based as the randomness is far more intermittent than in a DDPM where Gaussian noise would be added after every $1/\mathrm{poly}(d)$-sized step of the sampler. Nevertheless, this sampling algorithm is a significant deviation from how DDIMs work in practice due to the need for underdamped Langevin correction.

Under the same assumptions, Gupta et al. (2025) slightly improved the dimension dependence. Li & Jiao (2025) subsequently obtained dimension dependence of $O(Ld^{1/3}/\varepsilon^{2/3})$ by replacing the underdamped Langevin corrector with Gaussian noise, and with the same algorithmic template, recently Jiao & Li (2025) achieved $\min(d, L^{1/3}d^{2/3}, Ld^{1/3})/\varepsilon^{2/3}$. For the $L$-dependent part of their bound, they only require that the true score is locally Lipschitz with Lipschitz constant scaling similarly to our $L_t$. The main novelty of our result is that (1) we show the first sublinear bound for *SDEs*, which answers an open question posed by Jiao & Li (2025) about analyzing randomized midpoint for pure DDPM-based sampling, and (2) our algorithm simply runs the DDPM reverse process, without any corrector steps. For samplers that purely run the probability flow ODE without corrector steps, Li et al. (2024b); Huang et al. (2025) were the first to obtain polynomial convergence bounds without dependence on smoothness, though the best known dimension dependence in this setting is linear.

**Randomized midpoint method in sampling.** The randomized midpoint method was first introduced by Shen & Lee (2019) in the context of log-concave sampling with Langevin Monte Carlo.

A discussion of its use in that literature would take us too far afield, and we defer to the monograph of Chewi (2025) for details. We mention, however, that besides the shifted composition method that we apply, there is also a direct KL analysis of midpoint methods using anticipating Girsanov (Zhang, 2025), which however cannot achieve sharp rates. There is also a separate approach in Kandasamy & Nagaraj (2024); see Altschuler & Chewi (2024b) for comparisons and discussion.

In the context of diffusion models, the randomized midpoint method has been incorporated into all recent results on ODE-based sampling with sublinear complexity (Gupta et al., 2025; Jiao & Li, 2025; Li & Jiao, 2025). On the empirical front, Kandasamy & Nagaraj (2024); Gupta et al. (2025) provided experimental evidence for the favorable scaling of randomized midpoint for diffusion-based sampling.

## 2 PRELIMINARIES

**Notation.** We will use $\gamma$ to denote a standard Gaussian distribution over $\mathbb{R}^d$. The notation $a = O(b)$ or $a \lesssim b$ means that $a \leq cb$ for an absolute constant $c$ (i.e., not depending on the dimension, accuracy, or smoothness parameters), and similarly $a = \Omega(b), a \gtrsim b$ for $a \geq cb$. $a = \Theta(b)$ or $a \asymp b$ implies $a \lesssim b, a \gtrsim b$ simultaneously. Finally, the notation $\widetilde{O}, \widetilde{\Omega}, \widetilde{\Theta}$ means $O, \Omega, \Theta$ respectively up to extra polylogarithmic factors in $b$.

**Denoising diffusions.** We introduce the formalism of denoising diffusion probabilistic models (DDPMs). Let $\pi_0 \in \mathcal{P}(\mathbb{R}^d)$ denote the data distribution. The forward process is defined by evolving $\pi_0$ along the Ornstein–Uhlenbeck (OU) semigroup, which describes the SDE

$$\mathrm{d}X_t^{\rightarrow} = -X_t^{\rightarrow}\,\mathrm{d}t + \sqrt{2}\,\mathrm{d}B_t^{\rightarrow}\,, \qquad X_0^{\rightarrow} \sim \pi_0\,, \qquad \pi_t := \mathrm{law}(X_t^{\rightarrow})\,, \tag{OU}$$

where $(B_t)_{t \geq 0}$ is a standard Brownian motion. As is well-known by now, this equation admits a time-reversal (with respect to an initial measure $\pi_0$ and terminal time $T \in \mathbb{R}_+$) given by

$$\mathrm{d}X_t^{\leftarrow} = \left\{-X_t^{\leftarrow} + 2\,\nabla \log \frac{\pi_{T-t}}{\gamma}(X_t^{\leftarrow})\right\}\mathrm{d}t + \sqrt{2}\,\mathrm{d}B_t^{\leftarrow}\,, \tag{rev-OU}$$

where $(B_t^{\leftarrow})_{t \in [0,T]}$ is another standard Brownian motion. If (rev-OU) is initialized with $X_0^{\leftarrow} \sim \pi_T$, then $\mathrm{law}(X_t^{\leftarrow}) = \pi_{T-t}$ for all $t \in [0,T]$. As $\lim_{T \to \infty} \pi_T = \gamma$, we can view (OU) as a stochastic flow of $\pi_0$ to a standard Gaussian, and conversely (rev-OU) as a mechanism for obtaining samples from $\pi_0$ when starting from a standard Gaussian measure, assuming access to the score functions $(\nabla \log \pi_t)_{t \in [0,T]}$ or a suitable approximation. As we will generally be referring to (rev-OU) throughout this work, we will omit the $\cdot^{\leftarrow}$ in the notation with the reverse temporal direction being assumed.

**Algorithm.** Standard means for approximating (rev-OU) assume that the user has access to a process $(\mathsf{s}_t)_{t \in [0,T]}$ where $\mathsf{s}_t \approx \nabla \log \pi_t$ in a suitably strong sense. Simply substituting the estimator into (rev-OU) does not define a practical algorithm as the resulting SDE remains non-linear and hence does not admit a closed-form solution in general. Instead, one typically opts to discretize it by an appropriate linearization, for instance the exponential integrator given below. This solves the following SDE on $[t_k, t_{k+1})$ for a sequence of interpolant times $0 = t_0 < t_1 < t_2 < \ldots < t_N \leq T$:

$$\mathrm{d}X_t^{\mathrm{EE}} = \{-X_t^{\mathrm{EE}} + 2\widetilde{\mathsf{s}}_{T-t_k}(X_{t_k}^{\mathrm{EE}})\}\mathrm{d}t + \sqrt{2}\,\mathrm{d}B_t\,. \tag{EE}$$

For convenience, we have defined $\widetilde{\mathsf{s}}_t := \mathsf{s}_t - \nabla \log \gamma$. Conditional on $X_{t_k}^{\mathrm{EE}}$, this SDE is linear, so we can now compute an exact solution explicitly.

However, intuition from the field of log-concave sampling (Shen & Lee, 2019; Altschuler & Chewi, 2024b) suggests that a *randomized midpoint* discretization can significantly outperform the method above. Define a sequence of random variables $\tau_k$ with distribution function $f_k(\tau) = \frac{e^{\tau - h_k}}{1 - e^{-h_k}}$ over $[0, h_k]$. Then, the algorithm produces a sequence of iterates $X_{t_k}^{\mathsf{alg}}$ starting at $X_{t_0}^{\mathsf{alg}} = X_0^{\mathsf{alg}} \sim \gamma$, as follows: at step $k$ for $k \in [N]$, for $t \in [t_{k-1}, t_k)$,

$$X_t^+ := e^{-(t-t_{k-1})} X_{t_{k-1}}^{\mathsf{alg}} + 2\left(1 - e^{-(t-t_{k-1})}\right)\widetilde{\mathsf{s}}_{T-t_{k-1}}(X_{t_{k-1}}^{\mathsf{alg}}) + \sqrt{2}\int_{t_{k-1}}^t e^{s-t}\,\mathrm{d}B_s\,,$$

$$X_{t_k}^{\mathsf{alg}} := e^{-h_k} X_{t_{k-1}}^{\mathsf{alg}} + 2\left(1 - e^{-h_k}\right)\widetilde{\mathsf{s}}_{T-t_{k-1}-\tau_k}(X_{t_{k-1}+\tau_k}^+) + \sqrt{2}\int_{t_{k-1}}^{t_k} e^{s-t_k}\,\mathrm{d}B_s\,, \tag{RMD}$$

where $h_k := t_k - t_{k-1}$ is the step-size in the $k$-th iteration. Note that the two random variables

$$\xi_k^+ := \sqrt{2} \int_{t_{k-1}}^{t_{k-1}+\tau_k} e^{s-t_{k-1}-\tau_k} \, \mathrm{d}B_s \,, \qquad \xi_k := \sqrt{2} \int_{t_{k-1}}^{t_k} e^{s-t_k} \, \mathrm{d}B_s \,,$$

have an explicit distribution that can be easily simulated. See the lemma below.

**Lemma 2.** *For each $(\xi_k^+, \xi_k)$ defined above, we have*

$$\begin{bmatrix} \xi_k^+ \\ \xi_k \end{bmatrix} \sim \mathcal{N}\left( 0, \begin{bmatrix} 1 - e^{-2\tau_k} & e^{\tau_k - h_k} - e^{-(h_k+\tau_k)} \\ - & 1 - e^{-2h_k} \end{bmatrix} \otimes I_d \right),$$

*where the missing entry is determined by symmetry.*

The conditional means of (RMD) have simple closed forms, and so (RMD) corresponds to an easily computable Gaussian kernel.

---

**Algorithm 1:** Randomized midpoint kernel $P_k^{\mathsf{alg}}$ on $[t_{k-1}, t_k]$

---

**Input:** current state $X_{t_k}^{\mathsf{alg}} \in \mathbb{R}^d$; step $h_k := t_k - t_{k-1}$; score map $\mathsf{s}_t(\cdot)$.

**1. Draw the randomized midpoint**. Sample $U \sim \mathsf{Unif}(0,1)$ and set

$$\tau_k = h_k + \log\left(1 + U\left(e^{-h_k} - 1\right)\right) \quad \text{i.e., with density } f(\tau) = \frac{e^{\tau - h_k}}{1 - e^{-h_k}} \text{ on } [0, h_k].$$

**2. Midpoint prediction for $X_{t_k+\tau_k}^+$.** Draw $Z_1 \sim \mathcal{N}(0, I_d)$ and set the OU noise $\xi_k^+ := \sqrt{1 - e^{-2\tau_k}} \, Z_1$. Then

$$X_{t_{k-1}+\tau_k}^+ = e^{-\tau_k} X_{t_{k-1}}^{\mathsf{alg}} + 2\left(1 - e^{-\tau_k}\right) \widetilde{\mathsf{s}}_{T-t_{k-1}}(X_{t_{k-1}}^{\mathsf{alg}}) + \xi_k^+ \,.$$

**3. Full-step update for $X_{t_{k+1}}^{\mathsf{alg}}$.** Draw $Z_2 \sim \mathcal{N}(0, I_d)$ independent of $Z_1$ and set

$$\xi_k = e^{\tau_k - h_k} \xi_k^+ + \sqrt{1 - e^{2(\tau_k - h_k)}} \, Z_2 \,.$$

Compute the score at the randomized time and update

$$X_{t_k}^{\mathsf{alg}} = e^{-h_k} X_{t_{k-1}}^{\mathsf{alg}} + 2\left(1 - e^{-h_k}\right) \widetilde{\mathsf{s}}_{T-t_{k-1}-\tau_k}(X_{t_{k-1}+\tau_k}^+) + \xi_k \,.$$

---

## 3    RESULTS

We first delineate the assumptions underlying our results. We begin with two relatively benign conditions that are standard in the literature.

**Assumption 1** ($L^2$ accurate estimator)**.** Assume that for all $t \in [0, T]$, we have

$$\mathbb{E}_{\pi_t}[\|\nabla \log \pi_t - \mathsf{s}_t\|^2] \leq \varepsilon_{\text{score}}^2 \,.$$

**Assumption 2** (Bounded second moment)**.** Assume that the initial distribution has bounded second moment

$$\mathbb{E}_{\pi_0}[\|\cdot\|^2] \leq \mathsf{M}_2^2 < \infty \,.$$

**Assumption 3** (Time-varying smoothness)**.** For all $t \in [0, T]$, the estimated score has a Lipschitz constant bounded as follows: for all $x, y \in \mathbb{R}^d$,

$$\|\widetilde{\mathsf{s}}_t(x) - \widetilde{\mathsf{s}}_t(y)\| \leq \frac{\tilde{\beta}_0 \|x - y\|}{1 - e^{-2t}} \,.$$

As discussed in §1.1, these assumptions are a strict subset of those used in almost all existing works on diffusion-based sampling in sublinear complexity (Chen et al., 2023b; Gupta et al., 2025; Li & Jiao, 2025), with the exception of the recent work of Jiao & Li (2025) for which a weaker local Lipschitzness condition sufficed in place of Assumption 3. In Appendix B we provide examples of distributions for which the true scores are singular at time 0 (i.e., not Lipschitz uniformly in time), but which are covered by Assumption 3.

**Theorem 3** (Main result). *Suppose that Assumptions 1, 2, and 3 hold. Then Algorithm 1 with a decaying step size schedule can obtain a sample at time $t_N$ from a distribution $\hat{\pi}$ such that there is another distribution $\pi^{\mathsf{approx}}$ with*

$$\mathsf{KL}(\pi^{\mathsf{approx}} \parallel \hat{\pi}) \lesssim \widetilde{O}\big((1 + \log^2\{(1 \vee \tilde{\beta}_0)\,(d + \mathtt{M}_2^2)\})\,\varepsilon_{\mathrm{score}}^2\big)\,, \qquad W_2^2(\pi^{\mathsf{approx}}, \pi_0) \lesssim \varepsilon_{\mathrm{score}}^2\,,$$

*for $\varepsilon_{\mathrm{score}} \in (0, 1]$, $T \asymp \log \frac{d + \mathtt{M}_2^2}{\varepsilon_{\mathrm{score}}^2} \vee 1$ with no more than the following number of steps:*

$$N = \widetilde{\Theta}\Big(\frac{\tilde{\beta}_0 \sqrt{d + \mathtt{M}_2^2}}{\varepsilon_{\mathrm{score}}}\Big)\,.$$

**Remark.** In our analysis, we consider an algorithmic variant of (RMD) wherein each $\tau_k$ is not supported on $h_k$, but rather on a truncation $[0, \varrho_k h_k]$ where $1 - \varrho_k \ll 1$ is suitably small. This does not appreciably change the algorithm and is only done for technical convenience.

Although the nature of the guarantee may initially seem opaque (namely, the existence of an "intermediate" measure $\pi^{\mathsf{approx}}$), we note that standard results in the literature only guarantee TV (or KL) closeness to the early stopped distribution $\pi_\delta$ for some $\delta > 0$.[1] The usual justification for this is that $\pi_\delta$ is close to $\pi_0$ in $W_2$ distance, when $\delta$ is small. This early stopping assumption is so prevalent that it is often made with little fanfare, but we emphasize this point here to argue that our guarantee (KL-close-to-$W_2$-close) is of the same nature.[2] We remark, however, that $\pi^{\mathsf{approx}}$ is constructed from our proof technique and does not correspond to an early stopped distribution.

See Appendix A for more details on the step size schedules and proofs of the theorems.

## 4 TECHNICAL OVERVIEW

We first discuss the difficulties inherent in analyzing (RMD). The original analysis of Shen & Lee (2019), which inspired almost all subsequent analyses of randomized midpoint, is based on a coupling argument in $W_2$. However, **all state-of-the-art analyses for diffusion models work in TV or KL**. When we try to apply the former to the latter, we therefore arrive at a fundamental incongruity. Indeed, $W_2$ analyses of diffusion models often incur exponential accumulation of errors, unless overly restrictive assumptions such as strong log-concavity are imposed on $\pi_0$, e.g., Bruno et al. (2025); Gao et al. (2025); Gao & Zhu (2025); Yu & Yu (2025). One way in which existing works achieving sublinear complexity have circumvented this is to introduce *corrector steps* which periodically inject randomness into the dynamics to essentially convert $W_2$ bounds into KL bounds (Chen et al., 2023b; Gupta et al., 2025; Jiao & Li, 2025; Li & Jiao, 2025). This is an option that we cannot afford in this work, as our goal is to simply analyze a discretization of the vanilla DDPM reverse process without further algorithmic modifications.

In light of this, how can we analyze randomized midpoint discretization in TV or KL? There are two main challenges. The first is that standard approaches, such as Girsanov's theorem, do not readily apply to (RMD), because natural interpolations of (RMD) are not Markovian: the intermediate point $X^+_{t_{k-1}+\tau_k}$ "sees into the future" for times $t \le t_{k-1} + \tau_k$.

The second challenge is that the analysis should be fairly sharp in order to see a tangible benefit from (RMD). Indeed, the intuition behind (RMD) is that the use of a randomized step size to define $X^+_{t_{k-1}+\tau_k}$ effectively "debiases" the algorithm. This is formalized via the notions of weak and strong errors (Milstein & Tretyakov, 2021). Consider a single iteration on $[t_k, t_{k+1}]$, with the random variables $X^{\mathsf{alg}}_{t_{k+1}}$ and $X_{t_{k+1}}$ obtained by solving (RMD) and (rev-OU) respectively, from the same initial condition $X^{\mathsf{alg}}_{t_k} = X_{t_k} = x$. The weak and strong errors are defined as follows:

$$\text{Weak error:} \qquad \|\mathbb{E}\,X^{\mathsf{alg}}_{t_{k+1}} - \mathbb{E}\,X_{t_{k+1}}\| \le \mathcal{E}_{\mathrm{weak}}(x)\,,$$

$$\text{Strong error:} \qquad \|X^{\mathsf{alg}}_{t_{k+1}} - X_{t_{k+1}}\|_{L^2} \le \mathcal{E}_{\mathrm{strong}}(x)\,.$$

---

[1]Under Assumption 3, this is by necessity, since $\pi_0$ could have singular support in which case TV closeness to $\pi_0$ is not possible.

[2]Moreover, it is sufficient to metrize weak convergence. In particular, it controls the bounded Lipschitz distance; see, e.g., Chen et al. (2023c).

These two notions loosely capture the squared "bias" and "variance" of the discretization scheme at a single step. When the weak error is substantially smaller than the strong error—as is the case for (RMD)—then one can prove improved discretization bounds, basically because stochastic fluctuations cancel out à la the central limit theorem.[3] Unfortunately, as can be seen from the definitions, the weak and strong errors are most easily controlled via coupling methods, which are most easily handled in $W_2$.

In summary, we require an analysis framework that works in TV or KL, which is flexible enough to handle discretizations without Markovian interpolations, and which can witness the benefits of smaller weak errors.

**Shifted composition.** In the literature on log-concave sampling, in which the randomized midpoint discretization first arose, obtaining KL guarantees was also a longstanding challenge until very recently. A series of papers (Altschuler & Chewi, 2024a;b; Altschuler et al., 2025) has developed a new framework, known as *shifted composition*, which satisfies our desiderata above. We therefore aim to adapt it to the setting of diffusion models.

Briefly, the idea behind shifted composition is that in order to control the KL divergence between two processes (taken to be the algorithm and the "ideal" process it approximates), we can introduce a third process—called the auxiliary process—which is initialized at one of the two processes but is *shifted* to hit the second process at a terminal time. The hitting condition ensures that the KL divergence between the two original processes at the terminal time is controlled by the KL between the first process and the auxiliary process. Due to the definition of the auxiliary process, this latter KL can be controlled in terms of a distance recursion which incorporates the weak and strong errors.

**Adaptation to diffusion models.** Although shifted composition is well-suited for our needs, we stress that there are additional technical challenges in the diffusion model setting. Namely, under Assumption 3, the Lipschitz constant is changing with time; moreover, Theorem 3 uses a non-uniform step size schedule. Accommodating these complications requires an extension of the original shifted composition framework; see Appendix A for details.

## 5 EXPERIMENTS

In this section, we perform several experiments in image synthesis using pre-trained models from the EDM codebase (Karras et al., 2022) and the EDM2 evaluation (Karras et al., 2024) to validate and extend our theoretical predictions. We first conduct a baseline comparison demonstrating that Algorithm 1 outperforms Euler–Maruyama as well as the exponential Euler integrator applied to (rev-OU), consistent with the prediction of Theorem 3. We then highlight some of the design decisions that go into applying Algorithm 1 in practice, where state-of-the-art implementations use stochastic processes distinct from the OU process considered in the theoretical portion of our work. In this setting, we demonstrate how a tailored adaptation of DDRaM can outperform the Heun sampler introduced in (Karras et al., 2022) even for deterministic ODE sampling.

**Baseline comparison.** We first compare Algorithm 1 to two common baselines—a standard Euler–Maruyama sampler and the exponential Euler sampler. Specifically, Euler–Maruyama reads

$$X_{t_k} = (1 - h_k)X_{t_{k-1}} + 2h_k \left(\mathsf{s}_{T-t_{k-1}}(X_{t_{k-1}}) + X_{t_{k-1}}\right) + \sqrt{2h_k}\,\xi_k\,, \quad \xi_k \sim \gamma \text{ i.i.d.}\,, \quad \text{(EMD)}$$

where the factor $2X_{t_{k-1}}$ originates from the relative score to the standard Gaussian used in (rev-OU). Here, we write (EMD) in terms of the non-relative score because this is what is available as a pre-trained model. The exponential integrator is given by the analytic solution of (EE), which reads

$$X_{t_k} = e^{-h_k}X_{t_{k-1}} + 2\left(1 - e^{-2h_k}\right)\left(\mathsf{s}_{T-t_{k-1}}(X_{t_{k-1}}) + X_{t_{k-1}}\right) + \sqrt{1 - e^{-2h_k}}\,\xi_k\,,$$
$$\xi_k \sim \gamma \text{ i.i.d.}\,. \quad \text{(EED)}$$

We note that (EED) can be viewed as a subset of Algorithm 1 where we choose $\tau_k = h_k$ deterministically, and where we take $X_{t_{k-1}+\tau_k}^+$ as the next step $X_{t_k}$ without an intermediate.

---

[3]A simple analogy is that the sum of $N$ i.i.d. random variables, each with mean $\mu$ and standard deviation $\sigma$, has size roughly $N\mu + N^{1/2}\sigma$; think of $\mu$ as the weak error and $\sigma$ as the strong error.

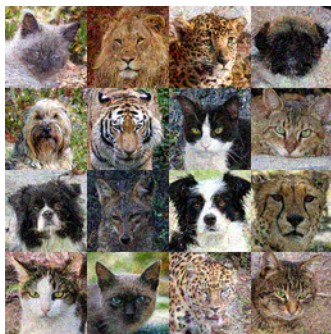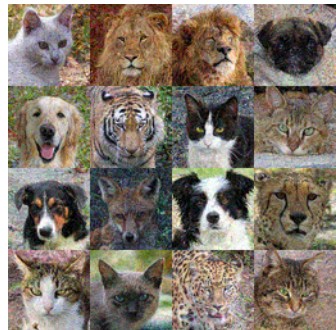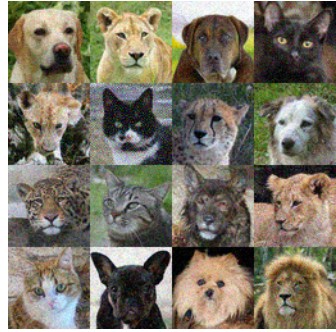

Figure 1: **Qualitative baseline comparison.** Listing from left to right, we show a qualitative comparison between the Euler–Maruyama sampler (EMD), the Euler exponential integrator (EED), and Algorithm 1 on the AFHQv2 dataset (Choi et al., 2020). All samplers use 64 score function evaluations (64 Euler integration steps, 32 midpoint steps) and leverage the EDM pre-trained unconditional VP model from Karras et al. (2022) at $64 \times 64$ resolution over the OU process (rev-OU). We fix the "VP" step size schedule of Song et al. (2021b) (Table 1 of Karras et al. (2022)) for all three methods. Clearly Algorithm 1 attains the best visual performance, which we quantify in Figure 2.

Results for the comparison between Algorithm 1, (EMD) and (EED) are shown in Figures 1 and 2 on the AFHQv2 dataset (Choi et al., 2020). Visually and quantitatively, Algorithm 1 performs best of the three.

**Beyond the OU process.** Although theoretical works uniformly analyze the OU process, practitioners often prefer time and space reparametrizations for both training and sampling. Examples of these include the "variance preserving" (VP) and "variance exploding" (VE) SDEs introduced by Song et al. (2021b), as well as the continuous limit of the DDPM schedule (Ho et al., 2020) suggested by Karras et al. (2022). It is *a priori* unclear how to adapt Algorithm 1 to these settings, though we may expect to attain similar practical gains to those seen on the OU process given a suitable extension. In order to extend to these new processes, we use a generalization of the key idea behind Algorithm 1 to handle a time-dependent scaling factor $\lambda(t)$, treating SDEs of the form

$$\mathrm{d}X_t = (\lambda(t)X_t + f_t(X_t))\,\mathrm{d}t + g(t)\,\mathrm{d}B_t\,. \quad \text{(SDE)}$$

In (SDE), we have the flexibility to choose $\lambda(t)$ by appropriate re-definition of $f_t$, which leads to a family of "randomized midpoint" methods parameterized by its choice. The resulting discretization scheme depends on various integrated quantities of $\lambda(t)$. For example, the choice $\lambda(t) = 0$ generates the randomized midpoint with Euler updates as opposed to the randomized midpoint with *exponential* Euler updates considered in Algorithm 1. Furthermore, when $g(t) = 0$, we notably recover a second-order ODE solver as a special case of the SDE solver. We provide further details in Appendix C.2.

Armed with this additional flexibility, we turn to the concrete setting of Karras et al. (2022), which considers a reparameterization of (SDE) of the form

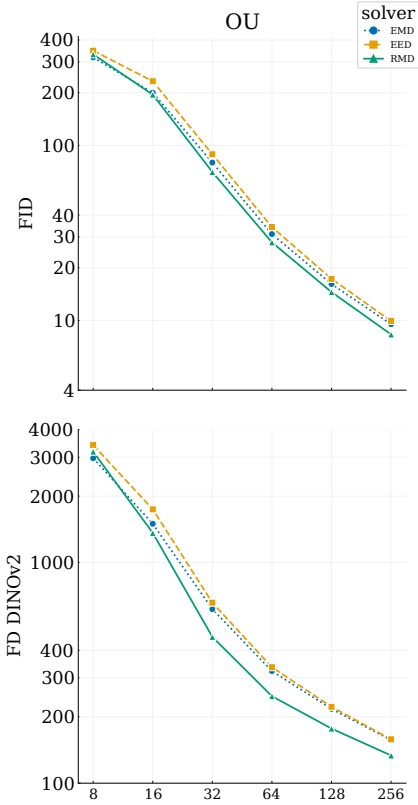

Figure 2: **Quantitative baseline comparison.** Image quality measured by FID (top) and FD$_{\text{DINOv2}}$ (bottom) versus number of score function evaluations (NFEs) for the (EMD), (EED), and (RMD) methods run on the OU process. Supporting Figure 1, (RMD) obtains the best quantitative results.

$$\mathrm{d}X_t = \left[\frac{\dot{c}(t)}{c(t)}X_t - (c(t)^2\dot{\sigma}(t)\sigma(t) + \beta(t)\sigma(t)^2c(t)^2)\,\hat{\mathbf{s}}_t(X_t)\right]\mathrm{d}t + \sqrt{2\beta(t)}\sigma(t)c(t)\,\mathrm{d}B_t\,, \quad \text{(EDM)}$$

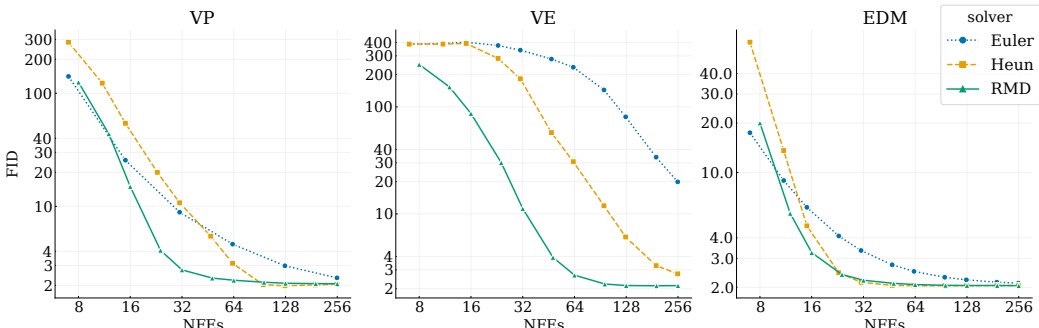

Figure 3: **Quantitative results: ODE sampling.** Image quality measured by Fréchet inception distance (FID↓) with number of score function evaluations (NFEs) for the Euler, Heun, and randomized midpoint methods. Columns correspond to the VP, VE, and EDM processes (and step size schedules). For $n$ steps of the solver, Euler takes $n$ NFEs, Heun takes $2n - 1$ (since Karras et al. (2022) run Euler on the last step to avoid the singularity at 0), and (RMD) takes $2n$. As a result, (RMD) has one more NFE than Euler and Heun in these plots. Figure 5 measures using $\text{FD}_{\text{DINOv2}}$ on the same images and shows similar results. Figure 6 shows the NFE curves on a shared $y$-axis.

where we have written $\hat{\mathsf{s}}_t(X_t) := \nabla \log \pi(X_t/c(t); \sigma(t))$ for the $(c, \sigma)$-parameterized score[4]. Note that the VP SDE, VE SDE, and OU process can all be recovered with appropriate choices of $c$ and $\sigma$.

We observe that for any choice of $c$ and $\sigma$, (EDM) will have: (1) a term with a time-dependent scaling of $X_t$; (2) a time-scaling of the score; and (3) a noise term which is a time-dependent scaling of the Wiener process (which is independent of $X_t$). In our experiments, we mainly consider two natural choices for $\lambda(t)$ such that the remaining drift of (EDM) is either a scaling of the score $\hat{\mathsf{s}}_t$ or the relative score $\widetilde{\mathsf{s}}_t$, with the time-dependent scaling of $X_t$ integrated exactly (see Appendix C.2.2). Our experiments suggest that $\lambda(t)$ should be chosen so the remaining drift is written entirely in terms of the score for the ODE and in terms of the relative score for the SDE. For our evaluations, we measure the Fréchet inception distance (FID) and Fréchet distance in the DINOv2 ($\text{FD}_{\text{DINOv2}}$) latent space (Oquab et al., 2024) as suggested by Stein et al. (2023); Karras et al. (2024) over a batch of 50k generated samples. Results are shown in Figure 3 and Figure 5, respectively.

We test deterministic sampling ($\beta(t) = 0$) on the AFHQv2 dataset (Choi et al., 2020) using the pre-trained VP model from Karras et al. (2022) over the VP, VE, and EDM processes. As shown in Figure 3 and Figure 5, we find that Algorithm 1 outperforms both the Euler and Heun samplers at essentially every NFE for all three settings considered in Karras et al. (2022), highlighting the advantages of DDRaM over widely-adopted diffusion solvers. We further note that DDRaM empirically seems to be far more robust to the choice of noise scheduler compared to Euler and Heun, where the NFE curves do not vary as much between processes. This is clearly seen in Figure 6.

## 6 CONCLUSION

In this paper, we have shown that stochastic diffusion model samplers can break the $O(d)$ complexity barrier given the right discretization and a natural Lipschitz assumption for the score estimator. Empirically, we find the randomized midpoint performs well in a variety of settings, outperforming both Euler and Heun for both stochastic SDE and deterministic ODE sampling. Several interesting lines of exploration remain for future work. First, it may be possible to combine our analysis with works that establish discretization guarantees depending only on the *intrinsic* dimension. Second, it would be quite interesting if Assumptions 3 on the score estimator could be removed, thereby providing an analysis under the minimal assumptions of Benton et al. (2024); Conforti et al. (2025a). This may be challenging, as it seems incompatible with our proof technique. Another possible avenue would be to replace our Lipschitz conditions with a relaxed Lipschitz condition, similarly to Jiao & Li (2025), which would imply substantially better guarantees for Gaussian mixtures.

---

[4]Karras et al. (2022) uses $s(t)$ for the scaling factor rather than $c(t)$. To avoid clash with our notation for the score, we opt to use $c$.

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

## A  DEFERRED PROOFS

### A.1  PRELIMINARY LEMMAS

Before proceeding, the following two generic lemmas will be useful in both regimes. Define

$$\mathsf{g}_t^2 := \mathbb{E}_{\pi_{T-t}}\big[\big\|\nabla \log \frac{\pi_{T-t}}{\gamma}\big\|^2\big].$$

We note that $\mathsf{g}_t^2 = \mathsf{FI}(\pi_{T-t} \,\|\, \gamma)$.

**Lemma 4** (Magic lemma I, adapted from Conforti et al. (2025a, Lemma 5)). *It holds that, letting* $\mathsf{M}_2^2 := \mathbb{E}_{\pi_0}[\|\cdot\|^2]$ *be the initial second moment,*

$$\mathsf{g}_t^2 \lesssim \frac{d}{1 - e^{-2(T-t)}} + \mathsf{M}_2^2.$$

The next lemma follows from a computation based on Itô's lemma.

**Lemma 5** (Magic lemma II, adapted from Conforti et al. (2025a, Proof of Lemma 2)). *It holds that, for* $s < t$, *letting* $\pi_{T-s,T-t}$ *be the joint law of the particle from (rev-OU) at times* $\{s, t\}$,

$$\mathbb{E}_{(X_s, X_t) \sim \pi_{T-s,T-t}}\big[\big\|\nabla \log \frac{\pi_{T-t}}{\gamma}(X_t) - \nabla \log \frac{\pi_{T-s}}{\gamma}(X_s)\big\|^2\big] \lesssim \mathsf{g}_t^2 - \mathsf{g}_s^2.$$

### A.2  REVIEW OF THE SHIFTED COMPOSITION FRAMEWORK

The shifted composition framework for proving discretization bounds for Markov processes, developed in the sequence of works Altschuler & Chewi (2024a;b); Altschuler et al. (2025), allows for the translation of Wasserstein/coupling-based errors to KL guarantees. They also allow the user to account for the difference between "weak" and "strong" errors. Suppose first that the following assumptions hold.

**Assumption 4** (Wasserstein regularity results). Suppose that we have a sequence of kernels $(P_n)_{n\geq0}$, $(P_n^{\mathsf{alg}})_{n\geq0}$ for which the following properties hold. Namely, for any $n \in \mathbb{N}$, let $x, y \in \mathbb{R}^d$, and let $X^{\mathsf{alg}} \sim \delta_x P_n^{\mathsf{alg}}$, $Y \sim \delta_y P_n$, $Y^{\mathsf{alg}} \sim \delta_y P_n^{\mathsf{alg}}$ be coupled. Then, for functions $\mathcal{E}_{\text{weak}}, \mathcal{E}_{\text{strong}} : \mathbb{R}^d \to \mathbb{R}_+$, constants $L, \gamma \geq 0$, assume that the following hold:

1. **Weak error:** $\|\mathbb{E}\, Y^{\mathsf{alg}} - \mathbb{E}\, Y\| \leq \mathcal{E}_{\text{weak}}(y)$.

2. **Strong error:** $\|Y^{\mathsf{alg}} - Y\|_{L^2} \leq \mathcal{E}_{\text{strong}}(y)$ for some coupling of $(Y^{\mathsf{alg}}, Y)$.

3. **Wasserstein Lipschitzness:** $\|X^{\mathsf{alg}} - Y^{\mathsf{alg}}\|_{L^2} \leq L \|x - y\|$ for some coupling of $(X^{\mathsf{alg}}, Y^{\mathsf{alg}})$.

4. **Coupling:** $\|X^{\mathsf{alg}} - x - (Y^{\mathsf{alg}} - y)\|_{L^2} \leq \gamma \|x - y\|$ for some coupling of $(X^{\mathsf{alg}}, Y^{\mathsf{alg}})$.

Without loss of generality in this work, assume $L \geq 1$.

Additionally, some conditions on the KL divergence are necessary to obtain guarantees.

**Assumption 5** (KL regularity). With the same notation as Assumption Assumption 4, assume that the following holds: for a parameter $c \geq 0$ and all $n \in \mathbb{N}$, $\mathsf{KL}(\delta_x P_n^{\mathsf{alg}} \parallel \delta_y P_n^{\mathsf{alg}}) \leq c \|x - y\|^2$.

Then, the following guarantee holds.

**Lemma 6.** *Under Assumptions Assumption 4 and Assumption 5, if $L \geq 1 + \frac{\log N}{N}$, we have for some measure $\pi^{\mathrm{approx}}$ and any initial measure $\pi \in \mathcal{P}(\mathbb{R}^d)$ that, defining $\bar{P}_k^{\mathsf{alg}} := P_1^{\mathsf{alg}} \cdots P_k^{\mathsf{alg}}$ and $\bar{P}_k := P_1 \cdots P_k$,*

$$\mathsf{KL}(\pi^{\mathrm{approx}} \parallel \pi \bar{P}_N^{\mathsf{alg}}) \lesssim c \left( \{(L-1)N \vee \log N\} \bar{\mathcal{E}}_{\mathrm{strong}}^2 + (N-1)\left(\bar{\mathcal{E}}_{\mathrm{weak}}^2 + \gamma \bar{\mathcal{E}}_{\mathrm{strong}}^2\right)\right).$$

*Furthermore, we have for $N \geq 2L/(L-1)$,*

$$W_2^2(\pi^{\mathrm{approx}}, \pi \bar{P}_N^{\mathsf{alg}}) \lesssim \bar{\mathcal{E}}_{\mathrm{strong}}^2 + \log\left(\frac{L}{L-1}\right)\left(\bar{\mathcal{E}}_{\mathrm{weak}}^2 + \gamma \bar{\mathcal{E}}_{\mathrm{strong}}^2\right).$$

*Here, $\bar{\mathcal{E}}_{\mathrm{strong}}^2 = \max_{k \in [N-1]} \mathbb{E}_{\mu \bar{P}_k}[\mathcal{E}_{\mathrm{strong}}^2]$ and $\bar{\mathcal{E}}_{\mathrm{weak}}^2$ is similarly defined.*

The proof of this theorem is accomplished by considering, if $(X_{t_k}^{\mathsf{alg}})_{k \in [N]}, (Y_{t_k})_{k \in [N]}$ are two processes which are started at the same random variable $X_0^{\mathsf{alg}} = Y_0$ and which evolve according to the kernels $(P_k^{\mathsf{alg}})_{k \in [N]}$ and $(P_k)_{k \in [N]}$ respectively, a third random variable

$$\tilde{Y}_0^{\mathsf{aux}} := Y_0, \qquad \tilde{Y}_{t_n}^{\mathsf{aux}} := Y_{t_n}^{\mathsf{aux}} + \eta_n\left(Y_{t_n} - Y_{t_n}^{\mathsf{aux}}\right), \qquad Y_{t_{n+1}}^{\mathsf{aux}} \sim P_{n+1}^{\mathsf{alg}}(\tilde{Y}_{t_n}^{\mathsf{aux}}, \cdot),$$

for an appropriate sequence of shifts $(\eta_n)_{n \in [N]}$, and then judiciously applying Assumptions Assumption 4 and Assumption 5. Note that the framework above does not account for the case where the constants vary between the different indices of the kernels $k \in \mathbb{N}$. This is the cause of substantial difficulties in our analysis, and will be focal point of our technical efforts.

### A.3 LOCAL ERROR ANALYSIS

We start by establishing local error estimates. When performing our analysis, we actually consider $\tau_k$ having the distribution function with density $f(\tau) \propto e^{\tau - h_k}$ for $\tau \in [0, \varrho_k h_k)$ for technical reasons. In practice, the choice of $\varrho_k \in (0, 1)$ makes little difference in the resulting bounds, and a more streamlined proof would not require such a truncation. We leave the clarification of this detail to future work. We define $(\tilde{\beta}_s)_{s \in [0,T]}$ to be an upper bound on the Lipschitz constant for $\tilde{\mathsf{s}}$, given by

$$\tilde{\beta}_t = \frac{\tilde{\beta}_0}{1 - e^{-2(T-t)}},$$

We assume throughout that $\tilde{\beta}_0 \geq 1$.

We first observe that (rev-OU) can be written

$$X_{t_k} = e^{-h_k} X_{t_{k-1}} + 2 \int_{t_{k-1}}^{t_k} e^{t - t_k} \nabla \log \frac{\pi_{T-t}}{\gamma}(X_t) \, \mathrm{d}t + \sqrt{2} \int_{t_{k-1}}^{t_k} e^{t - t_k} \, \mathrm{d}B_t.$$

**Lemma 7** (Pointwise local errors). *Consider a fixed iteration $k \in [N]$. Under our previous assumptions, we have the following weak error bound, where $X_{t_k}^{\mathsf{alg}}(x)$ is from (RMD) with truncation, and $X_{t_k}(x)$ from (rev-OU), conditional on $X_{t_{k-1}}^{\mathsf{alg}} = X_{t_{k-1}} = x$ and solving both equations over $t \in [t_{k-1}, t_k)$ for $h_k \ll 1$:*

$$\|\mathbb{E}\, X_{t_k}^{\mathsf{alg}}(x) - \mathbb{E}\, X_{t_k}(x)\|^2 \lesssim h_k \int_{t_{k-1}}^{t_k} \left(\mathsf{F}_t^2(x) + \tilde{\beta}_{t_k}^2 h_k \int_{t_{k-1}}^t \left(\mathsf{G}_{t_{k-1},s}^2(x) + \mathsf{F}_{t_{k-1}}^2(x)\right) \mathrm{d}s\right) \mathrm{d}t$$

$$+ (1 - \varrho_k)^2 h_k^2 \sup_{t \in [t_{k-1}, t_k]} \mathbb{E}\|\tilde{\mathsf{s}}_{T-t}(X_t^+(x))\|^2,$$

*where $\mathsf{G}_{s,t}(x), \mathsf{F}_t(x)$ are defined as*

$$\mathsf{G}_{s,t}^2(x) := \mathbb{E}\left[\left\|\nabla \log \frac{\pi_{T-t}}{\gamma}(X_t(x)) - \nabla \log \frac{\pi_{T-s}}{\gamma}(X_s(x))\right\|^2\right],$$

$$\mathsf{F}_t^2(x) := \mathbb{E}\left[\|\mathsf{s}_{T-t}(X_t(x)) - \nabla \log \pi_{T-t}(X_t(x))\|^2\right],$$

*for $t \in [t_{k-1}, t_k]$, starting from $X_{t_{k-1}} = X_{t_{k-1}}^{\mathsf{alg}} = x$. Also,*

$$\mathbb{E}[\|X_{t_k}^{\mathsf{alg}}(x) - X_{t_k}(x)\|^2] \lesssim h_k^2 \left( \mathbb{E}\, \mathsf{F}_{t_{k-1}+\tau_k}^2(x) + \tilde{\beta}_{t_k}^2 h_k\, \mathbb{E} \int_{t_{k-1}}^{t_{k-1}+\tau_k} (\mathsf{G}_{t_{k-1},s}^2(x) + \mathsf{F}_{t_{k-1}}^2(x))\, \mathrm{d}s \right)$$

$$+ h_k \int_{t_{k-1}}^{t_k} \mathbb{E}\, \mathsf{G}_{t_{k-1}+\tau_k,t}^2(x)\, \mathrm{d}t\,.$$

**Proof.** We will suppress the argument in $X_t^{\mathsf{alg}}(x)$, $X_t(x)$, $X_t^+(x)$, considering always a fixed starting point $x$. Note that for $h_k \ll 1$,

$$\left| \frac{e^{t-h_k}}{1 - e^{-h_k}} - \frac{e^{t-h_k}}{\int_0^{\varrho_k h_k} e^{s - h_k}\, \mathrm{d}s} \right| \lesssim \frac{1 - \varrho_k}{\varrho_k} \cdot \frac{1}{h_k}\,,$$

for $t \in [0, \varrho_k h_k)$. On the other hand, the maximum of $\frac{e^{t-h_k}}{1 - e^{-h_k}}$ on $[\varrho_k h_k, h_k)$ is bounded by at most a constant times $h_k^{-1}$. It follows that, taking $X_{t_k}^{\mathsf{untrc}}$ from (RMD) without truncation of the distribution for $\tau_k$, that

$$\|\mathbb{E}\, X_{t_k}^{\mathsf{alg}} - \mathbb{E}\, X_{t_k}\| \le \|\mathbb{E}\, X_{t_k}^{\mathsf{untrc}} - \mathbb{E}\, X_{t_k}^{\mathsf{alg}}\| + \|\mathbb{E}\, X_{t_k}^{\mathsf{untrc}} - \mathbb{E}\, X_{t_k}\|$$

$$= \left\| 2\,(1 - e^{-h_k})\, \mathbb{E} \int_0^{h_k} \widetilde{\mathsf{s}}_{T - t_{k-1} - \tau}(X_{t_{k-1}+\tau}^+) \left( \frac{e^{\tau - h_k}}{1 - e^{-h_k}} - \frac{e^{\tau - h_k}\, \mathbb{1}_{\tau \le \varrho_k h_k}}{\int_0^{\varrho_k h_k} e^{\tau - h_k}\, \mathrm{d}\tau} \right) \mathrm{d}\tau \right\|$$

$$+ \left\| 2\, \mathbb{E} \int_0^{h_k} \left( \widetilde{\mathsf{s}}_{T - t_{k-1} + \tau}(X_{t_{k-1}+\tau}^+) - \nabla \log \frac{\pi_{T - t_{k-1} - \tau}}{\gamma}(X_{t_{k-1}+\tau}) \right) e^{\tau - h_k}\, \mathrm{d}\tau \right\|$$

$$\lesssim \frac{1 - \varrho_k}{\varrho_k} \int_0^{\varrho_k h_k} \mathbb{E}\|\widetilde{\mathsf{s}}_{T - t_{k-1} - t}(X_{t_{k-1}+t}^+)\|\, \mathrm{d}t + \int_{\varrho_k h_k}^{h_k} \mathbb{E}\|\widetilde{\mathsf{s}}_{T - t_{k-1} - t}(X_{t_{k-1}+t}^+)\|\, \mathrm{d}t$$

$$+ \int_{t_{k-1}}^{t_k} \mathbb{E}\|\mathsf{s}_{T-t}(X_t) - \nabla \log \pi_{T-t}(X_t)\|\, \mathrm{d}t + \tilde{\beta}_{t_k} \int_{t_{k-1}}^{t_k} \mathbb{E}\|X_t^+ - X_t\|\, \mathrm{d}t\,.$$

Now, we have

$$\mathbb{E}[\|X_t^+ - X_t\|^2] = \mathbb{E}\left[ \left\| \int_{t_{k-1}}^t \left( \widetilde{\mathsf{s}}_{T - t_{k-1}}(x) - \nabla \log \frac{\pi_{T-s}}{\gamma}(X_s) \right) e^{s - t_k}\, \mathrm{d}s \right\|^2 \right]$$

$$\lesssim h_k \int_{t_{k-1}}^t \Big( \mathbb{E}[\|\mathsf{s}_{T - t_{k-1}}(x) - \nabla \log \pi_{T - t_{k-1}}(x)\|^2] \tag{A.1}$$

$$+ \mathbb{E}\left[ \left\| \nabla \log \frac{\pi_{T-s}}{\gamma}(X_s) - \nabla \log \frac{\pi_{T - t_{k-1}}}{\gamma}(x) \right\|^2 \right] \Big)\, \mathrm{d}s\,.$$

On the other hand,

$$\|X_{t_k}^{\mathsf{alg}} - X_{t_k}\|^2 \lesssim \left\| \int_{t_{k-1}}^{t_k} \left( \widetilde{\mathsf{s}}_{T - t_{k-1} - \tau_k}(X_{t_{k-1}+\tau_k}^+) - \nabla \log \frac{\pi_{T-t}}{\gamma}(X_t) \right) e^{t - t_k}\, \mathrm{d}t \right\|^2$$

$$\lesssim h_k \int_{t_{k-1}}^{t_k} \|\nabla \log \pi_{T-t}(X_t) - \mathsf{s}_{T - t_{k-1} - \tau_k}(X_{t_{k-1}+\tau_k})\|^2\, \mathrm{d}t$$

$$+ \tilde{\beta}_{t_k}^2 h_k^2\, \|X_{t_{k-1}+\tau_k}^+ - X_{t_{k-1}+\tau_k}\|^2\,.$$

We then split the first term into

$$\mathbb{E}[\|\nabla \log \pi_{T-t}(X_t) - \mathsf{s}_{T - t_{k-1} - \tau_k}(X_{t_{k-1}+\tau_k})\|^2] \lesssim \mathsf{G}_{t_{k-1}+\tau_k,t}^2(x) + \mathsf{F}_{t_{k-1}+\tau_k}^2(x)\,.$$

For the second term, we can reuse (A.1). This gives our desired bound. $\qquad \square$

The following lemma follows from applying the Lipschitz property of the estimator, as well as the bound (A.1) that we previously derived.

**Lemma 8** (Score estimator bounds). *We have, for $X_t^+(x)$ obtained from (RMD) conditional on $X_{t_{k-1}}^{\mathrm{alg}} = x$, for $t \in [t_{k-1}, t_k]$,*

$$\mathbb{E}[\|\widetilde{\mathsf{s}}_{T-t}(X_t^+(x))\|^2] \lesssim \tilde{\beta}_{t_k}^2 h_k \int_{t_{k-1}}^t \left( \mathsf{G}_{t_{k-1},s}^2(x) + \mathsf{F}_{t_{k-1}}^2(x) \right) \mathrm{d}s$$

$$+ \mathsf{G}_{t_{k-1},t}^2(x) + \mathsf{F}_t^2(x) + \left\| \nabla \log \frac{\pi_{T-t_{k-1}}}{\gamma}(x) \right\|^2.$$

Recall that the local errors $(\bar{\mathcal{E}}_k^{\mathrm{weak}})^2$, $(\bar{\mathcal{E}}_k^{\mathrm{strong}})^2$ are simply the pointwise local errors from Lemma Lemma 7, averaged over $x \sim \pi_{T-t_{k-1}}$.

**Lemma 9** (Local errors). *For all $k \in \mathbb{N}$, we have the following errors, taking $1 - \varrho_k \asymp h_k^r$ for some power $r \geq 2$ at each step (treated as an absolute constant), with $h_k \ll 1/\tilde{\beta}_{t_k}$ always,*

*(a)* **Weak error:**

$$(\bar{\mathcal{E}}_k^{\mathrm{weak}})^2 \lesssim h_k^2 \varepsilon_{\mathrm{score}}^2 + \tilde{\beta}_{t_k}^2 h_k^4 (\mathsf{g}_{t_k}^2 - \mathsf{g}_{t_{k-1}}^2) + h_k^{2+2r} \mathsf{g}_{t_k}^2.$$

*(b)* **Strong error:**

$$(\bar{\mathcal{E}}_k^{\mathrm{strong}})^2 \lesssim h_k^2 \varepsilon_{\mathrm{score}}^2 + h_k^2 (\mathsf{g}_{t_k}^2 - \mathsf{g}_{t_{k-1}}^2).$$

*Note that the main difference between the two errors is the additional error term $h_k^2 (\mathsf{g}_{t_k}^2 - \mathsf{g}_{t_{k-1}}^2)$ in the strong error.*

**Proof.** To bound these in expectation, assuming that $X \sim \pi_{T-t_{k-1}}$, we have from Lemma Lemma 5,

$$\sup_{t_{k-1} \leq s \leq t \leq t_k} \mathbb{E}_{X \sim \pi_{T-t_{k-1}}}[\mathsf{G}_{s,t}^2(X)] \leq \mathsf{g}_{t_k}^2 - \mathsf{g}_{t_{k-1}}^2.$$

Here, we note that $t \mapsto \mathsf{g}_t^2$ is monotonically increasing along the Ornstein–Uhlenbeck semigroup.

On the other hand,

$$\sup_{t \in [t_{k-1}, t_k]} \mathbb{E}_{X \sim \pi_{T-t_{k-1}}}[\mathsf{F}_t^2(X)] \lesssim \varepsilon_{\mathrm{score}}^2.$$

Substituting these into Lemma Lemma 7, and using Lemma Lemma 8 concludes the proof. $\qquad\square$

A.4  VERIFYING THE ASSUMPTIONS OF SHIFTED COMPOSITION

Next, we check the hypotheses of the shifted composition local error framework (see Appendix Appendix A.2).

**Lemma 10** (Properties of (RMD)). *For all $k \in \mathbb{N}$, the Markov kernels $P_k^{\mathrm{alg}}$ corresponding to (RMD) satisfy the following properties, with the same definitions as Lemma Lemma 7. Let $Y^{\mathrm{alg}}$ denote the output of (RMD) starting from $y$. Assume that $h_k \ll 1/\tilde{\beta}_{t_k}$, and define $\mathsf{p}_k := \tilde{\beta}_{t_k} h_k$.*

*(a)* **Wasserstein Lipschitzness:** $\|X^{\mathrm{alg}} - Y^{\mathrm{alg}}\|_{L^2} - \|x - y\| \lesssim \mathsf{p}_k \|x - y\|$.

*(b)* **Coupling:** $\|X^{\mathrm{alg}} - Y^{\mathrm{alg}} - (x - y)\|_{L^2} \lesssim \mathsf{p}_k \|x - y\|$.

*(c)* **Regularity:** *Let $\varrho_k \in [0, 1)$ be a parameter which is arbitrarily close to 1. Then, we have*

$$\mathsf{KL}(\delta_x P_k^{\mathrm{alg}} \| \delta_y P_k^{\mathrm{alg}}) \lesssim \frac{\|x - y\|^2}{h_k} \log \frac{1}{1 - \varrho_k}.$$

**Proof.**

(a) This follows from (b).

(b) Fixing $\tau_k$ and synchronously coupling the Brownian motions, we have

$$\|X^{\mathsf{alg}} - Y^{\mathsf{alg}} - (x - y)\| \leq (1 - e^{-h_k}) \, \|x - y\|$$
$$+ 2 \, (1 - e^{-h_k}) \, \|\widetilde{\mathsf{s}}_{T-t_{k-1}-\tau_k}(X^+_{t_{k-1}+\tau_k}) - \widetilde{\mathsf{s}}_{T-t_{k-1}-\tau_k}(Y^+_{t_{k-1}})\|$$
$$\leq (1 - e^{-h_k}) \, \|x - y\| + 2\tilde{\beta}_{t_k} \, (1 - e^{-h_k}) \, \|X^+_{t_{k-1}+\tau_k} - Y^+_{t_{k-1}+\tau_k}\| \, .$$

As for the second term, we can bound it again via synchronous coupling:

$$\|X^+_{t_{k-1}+\tau_k} - Y^+_{t_{k-1}+\tau_k}\| = \|e^{-\tau_k} \, (x - y) + 2 \, (1 - e^{-\tau_k}) \, (\widetilde{\mathsf{s}}_{T-t_{k-1}}(x) - \widetilde{\mathsf{s}}_{T-t_{k-1}}(y))\|$$
$$\lesssim (1 + \tilde{\beta}_{t_k} h_k) \, \|x - y\| \lesssim \|x - y\| \, .$$

(c) We apply a familiar trick from Altschuler & Chewi (2024b) where we compute the conditional KL given $\tau_k$, and then integrate. It is for this reason that we need to truncate our random variable $\tau_k$. Condition on $\omega_k := \{\tau_k, (B_t)_{t \leq t_{k-1}+\tau_k}\}$. Then, we have

$$\delta_x P^{\mathsf{alg}}_{k|\omega_k} = \mathcal{N}\big(e^{-h_k}x + 2 \, (1 - e^{-h_k}) \, \widetilde{\mathsf{s}}_{T-t_{k-1}-\tau_k}(X^+_{t_{k-1}+\tau_k}) + \zeta_{k,1}, \, (1 - e^{-2(h_k - \tau_k)}) \, I_d\big) \, ,$$

where

$$\zeta_{k,1} = \sqrt{2} \int_{t_{k-1}}^{t_{k-1}+\tau_k} e^{s-t_k} \, \mathrm{d}B_s \, .$$

Using the formula for the KL divergence between two Gaussians, we find

$$\frac{\|e^{-h_k} \, (x - y) + 2 \, (1 - e^{-h_k}) \, (\widetilde{\mathsf{s}}_{T-t_{k-1}-\tau_k}(X^+_{t_{k-1}+\tau_k}) - \widetilde{\mathsf{s}}_{T-t_{k-1}-\tau_k}(Y^+_{t_{k-1}+\tau_k}))\|^2}{2 \, (1 - e^{-2(h_k - \tau_k)})}$$
$$\lesssim \frac{1}{1 - e^{-2(h_k - \tau_k)}} \, \|x - y\|^2 + \frac{\beta^2_{t_k} h^2_k}{1 - e^{-2(h_k - \tau_k)}} \, \|X^+_{t_{k-1}+\tau_k} - Y^+_{t_{k-1}+\tau_k}\|^2$$
$$\lesssim \frac{\|x - y\|^2}{1 - e^{-2(h_k - \tau_k)}} \, .$$

Linearizing the denominator for $h_k \lesssim 1$ and $\tau_k \in [0, \varrho_k h_k]$ for some parameter $\varrho_k$ approaching 1,

$$\mathsf{KL}(\delta_x P^{\mathsf{alg}}_{k|\omega_k} \, \| \, \delta_y P^{\mathsf{alg}}_{k|\omega_k}) \lesssim \frac{\|x - y\|^2}{h_k - \tau_k} \, .$$

Taking expectations and using joint convexity, we find

$$\mathsf{KL}(\delta_x P^{\mathsf{alg}}_k \, \| \, \delta_y P^{\mathsf{alg}}_k) \lesssim \frac{\|x - y\|^2}{h_k} \log \frac{1}{1 - \varrho_k} \, .$$

$\square$

## A.5 INTEGRAL COMPUTATIONS

Now, we need a bespoke version of the original local error recursion from Altschuler & Chewi (2024b) which holds for the time-varying step sizes considered in this work. We consider the following step size choice, which satisfies $h_k \ll 1/\tilde{\beta}_{t_k}$.

$$h_k := \frac{C_h \varepsilon_{\mathsf{score}}}{\tilde{\beta}_0 \, \sqrt{(d + \mathtt{M}^2_2) \, T}} \min\{1, T - t_k\} \asymp \frac{\varepsilon_{\mathsf{score}}}{\tilde{\beta}_0 \sqrt{(d + \mathtt{M}^2_2) \, T}} \cdot (1 - e^{-2(T - t_k)}) \, . \quad (\text{A.2})$$

Here, $C_h \asymp 1$ is an absolute constant. Let us briefly justify this. When $T - t_k \leq 1/\tilde{\beta}_0$, then $\frac{1}{1 - e^{-2(T-t_k)}} \asymp \frac{1}{T - t_k}$. Otherwise, $\frac{1}{1 - e^{-2(T-t_k)}} \asymp 1$. We also select the shift

$$\eta_t = \frac{C_\eta \tilde{\beta}_0}{1 - e^{-2(T-t)}} \, ,$$

where again $C_\eta \asymp 1$.

The following proof is heavily based on the argument of Altschuler & Chewi (2024b). Although we briefly describe the high-level idea in the subsequent proof, a detailed discussion of the shifted composition framework is beyond the scope of this paper and we refer to Altschuler & Chewi (2024b).

**Lemma 11.** *Under Assumptions Assumption 1, Assumption 2, and Assumption 3, with the choice of step-size given in (A.2) and for $T \geq 1$ and $t_N \in (T - \frac{1}{6}, T)$, there exists a probability measure $\pi_{t_N}^{\mathsf{aux}}$ such that*

$$\mathsf{KL}(\pi_{t_N}^{\mathsf{aux}} \parallel \pi_{t_N}^{\mathsf{alg}}) \lesssim \mathsf{KL}(\pi_T \parallel \gamma) + \left( T + \frac{1}{T} \log \frac{1}{T - t_N} \right) \varepsilon_{\mathrm{score}}^2 \log \frac{\tilde{\beta}_0 \sqrt{(d + \mathsf{M}_2^2) \, T}}{\varepsilon_{\mathrm{score}} \, (T - t_N)} \, .$$

*Furthermore, if we consider $d_N^2 = \|Y_{t_N}^{\mathsf{aux}} - Y_{t_N}\|_{L^2}^2$ where $Y_{t_N}^{\mathsf{aux}} \sim \pi_{t_N}^{\mathsf{aux}}$ and $Y_{t_N} \sim \pi_{t_N}$, then*

$$d_N^2 \lesssim \left( (T - t_N)^2 + \frac{T - t_N}{T} \right) \frac{\varepsilon_{\mathrm{score}}^2}{\tilde{\beta}_0^2} \, .$$

**Proof.** The idea is to define an auxiliary process $(Y_{t_n}^{\mathsf{aux}})_{n \leq N}$ with $Y_{t_N}^{\mathsf{aux}} \sim \pi_{t_N}^{\mathsf{aux}}$. The auxiliary process is defined as follows:

$$\tilde{Y}_0^{\mathsf{aux}} \sim \pi_T \, , \qquad \tilde{Y}_{t_n}^{\mathsf{aux}} := Y_{t_n}^{\mathsf{aux}} + \eta_n \left( Y_{t_n} - Y_{t_n}^{\mathsf{aux}} \right) \, , \qquad Y_{t_{n+1}}^{\mathsf{aux}} \sim P_{n+1}^{\mathsf{alg}}(\tilde{Y}_{t_n}^{\mathsf{aux}}, \cdot) \, .$$

Here, $\eta_n := \int_{t_{n-1}}^{t_n} \eta_t \, \mathrm{d}t$, and $(Y_t)_{t \in [0,T]}$ denotes (rev-OU). In other words, the auxiliary process follows the (RMD) algorithm (i.e., using an estimated score and time discretization), but we interleave steps which shift the auxiliary process toward the true reverse process.

By the KL chain rule,

$$\mathsf{KL}(\pi_{t_N}^{\mathsf{aux}} \parallel \pi_{t_N}^{\mathsf{alg}}) \leq \mathsf{KL}(\pi_T \parallel \gamma) + \mathbb{E}_{x \sim \pi_T} \mathsf{KL}(\mathbf{P}_x^{\mathsf{aux}} \parallel \mathbf{P}_x^{\mathsf{alg}}) \, ,$$

where $\mathbf{P}_x^{\mathsf{aux}}, \mathbf{P}_x^{\mathsf{alg}}$ denote path measures started from $x$.

Define $d_n^2 := \mathbb{E}[\|Y_n^{\mathsf{aux}} - Y_{t_n}\|^2]$ and note that $d_0 = 0$. We compute the KL divergence between the auxiliary process and the algorithm using the shifted composition technique and Lemma Lemma 10; see Altschuler & Chewi (2024b, §3).

$$\mathbb{E}_{x \sim \pi_T} \mathsf{KL}(\mathbf{P}_x^{\mathsf{aux}} \parallel \mathbf{P}_x^{\mathsf{alg}}) \lesssim \sum_{k=1}^{N} \frac{\eta_k^2 d_k^2}{h_k} \log \frac{1}{h_k} \lesssim \sum_{k=1}^{N} h_k \eta_{kh}^2 d_k^2 \log \frac{1}{h_k} \, .$$

The next step is to simplify the computation by approximating the sum by an integral, as was done in Altschuler et al. (2025). In this proof, we reserve the `mathtt` font for continuous-time interpolations of discrete quantities appearing in this proof. Thus, $\mathsf{d}_t^2$ interpolates $d_n^2$, i.e., $\mathsf{d}_t^2 := d_{t_n}^2$ where $t_n \leq t \leq t_{n+1}$. Similarly, $\mathsf{h}_t$ is defined similarly to $h_k$ in (A.2), replacing $t_k$ with $t$. Then,

$$\mathbb{E}_{x \sim \pi_T} \mathsf{KL}(\mathbf{P}_x^{\mathsf{aux}} \parallel \mathbf{P}_x^{\mathsf{alg}}) \lesssim \int_0^{t_N} \eta_t^2 \mathsf{d}_t^2 \log \frac{1}{\mathsf{h}_t} \, \mathrm{d}t \, .$$

We next write down a recursion for $d_n^2$. This is the usual local error recursion, see Altschuler & Chewi (2024b, Lemma B.5).

$$d_n^2 \leq (1 + \mathsf{p}_n)^2 \, (1 - \eta_n)^2 \, d_{n-1}^2 + 2 \left( \bar{\mathcal{E}}_n^{\mathrm{weak}} + \mathsf{p}_n \bar{\mathcal{E}}_n^{\mathrm{strong}} \right) (1 - \eta_n) \, d_{n-1} + (\bar{\mathcal{E}}_n^{\mathrm{strong}})^2 \, .$$

Here, we invoke Lemma Lemma 10, the conclusion of which involves hidden universal constants. By redefining $\mathsf{p}_n$ (so that $\mathsf{p}_n = O(\tilde{\beta}_{t_n} h_n)$), we write the above recursion without any universal constants, which simplifies the following computations.

Applying Young's inequality on the middle term, we find that

$$d_n^2 \leq (1 + \mathsf{p}_n) \, (1 - \eta_n) \, d_{n-1}^2 + O\left( \frac{(\bar{\mathcal{E}}_n^{\mathrm{weak}} + \mathsf{p}_n \bar{\mathcal{E}}_n^{\mathrm{strong}})^2}{(1 + \mathsf{p}_n) \, (1 - \eta_n) - (1 + \mathsf{p}_n)^2 \, (1 - \eta_n)^2} + (\bar{\mathcal{E}}_n^{\mathrm{strong}})^2 \right) \, .$$

To simplify the denominator, let us make the ansatz (which we will verify later) that $\mathsf{p}_n, \eta_n \ll 1$ and $(1 + \mathsf{p}_n) \, (1 - \eta_n) < 1$. This then yields the following recursion, noting that $d_0^2 = 0$ is assumed:

$$d_n^2 \lesssim \sum_{k=1}^{n} \left( \prod_{j=k+1}^{n} (1 + \mathsf{p}_j) \, (1 - \eta_j) \right) \left( \frac{(\bar{\mathcal{E}}_k^{\mathrm{weak}})^2}{\eta_k - \mathsf{p}_k} + (\bar{\mathcal{E}}_k^{\mathrm{strong}})^2 \right) \, .$$

In such a case, given our choice of step size and shift, defining

$$\mathbf{p}_t \asymp \frac{\tilde{\beta}_0 \mathbf{h}_t}{1 - e^{-2(T-t_k)}}, \qquad \mathbf{h}_t := \frac{C_h \varepsilon_{\text{score}}\left(1 - e^{-2(T-t)}\right)}{\tilde{\beta}_0 \sqrt{(d + \mathbf{M}_2^2)T}},$$

so that naturally $\mathbf{p}_k = \mathbf{p}_{t_k}$, $h_k = \mathbf{h}_{t_k}$, we can write

$$\eta_k - \mathbf{p}_k \asymp \frac{\tilde{\beta}_0 \mathbf{h}_{t_k}}{1 - e^{-2(T-t_k)}} \asymp \frac{\varepsilon_{\text{score}}}{\sqrt{(d + \mathbf{M}_2^2)T}},$$

under our choices as well. We indeed have $(1 + \mathbf{p}_n)(1 - \eta_n) < 1$ if we choose $C_\eta$ to be a sufficiently large absolute constant, and $\mathbf{p}_n, \eta_n \ll 1$.

Furthermore, define the following:

$$(\mathbf{E}_t^{\text{strong}})^2 := \varepsilon_{\text{score}}^2 \frac{\varepsilon_{\text{score}}\left(1 - e^{-2(T-t)}\right)}{\tilde{\beta}_0 \sqrt{(d + \mathbf{M}_2^2)T}} + \frac{\varepsilon_{\text{score}}^2\left(1 - e^{-2(T-t)}\right)^2}{\tilde{\beta}_0^2 (d + \mathbf{M}_2^2)T}\, \partial_t \mathbf{g}_t^2,$$

$$(\mathbf{E}_t^{\text{weak}})^2 := \varepsilon_{\text{score}}^2 \frac{\varepsilon_{\text{score}}\left(1 - e^{-2(T-t)}\right)}{\tilde{\beta}_0 \sqrt{(d + \mathbf{M}_2^2)T}} + \frac{\varepsilon_{\text{score}}^4\left(1 - e^{-2(T-t)}\right)^2}{\tilde{\beta}_0^2 (d + \mathbf{M}_2^2)^2 T^2}\, \partial_t \mathbf{g}_t^2.$$

These are obtained by taking the local errors from Lemma Lemma 9, dividing by a factor of $h_k$ (which is helpful when converting from the summation to the integral approximation), and taking the continuous-time interpolation. Here, the contribution from the $h_k^{2r}$ term can be seen to be negligible, taking $r \geq 4$ sufficiently large and bounding $\mathbf{g}^2$ using Lemma Lemma 4. Note that the finite difference $\mathbf{g}_{t_k}^2 - \mathbf{g}_{t_{k-1}}^2$ converts into a derivative. Finally, we have for absolute constants $\bar{c}, c$ that

$$\prod_{j=k+1}^{n} (1 + \mathbf{p}_j)(1 - \eta_j) \leq \exp\Big( \sum_{j=k+1}^{n} \Big(c\tilde{\beta}_{t_j} h_j - \frac{C_\eta \tilde{\beta}_0 h_j}{c\left(1 - e^{-2(T-t_j)}\right)}\Big)\Big)$$

$$\leq \exp\Big(-\int_{t_{k+1}}^{t_n} \frac{\bar{c}C_\eta \tilde{\beta}_0}{1 - e^{-2(T-t)}}\, \mathrm{d}t\Big),$$

so long as we choose $C_\eta$ to be a sufficiently large absolute constant. We then substitute this into

$$\mathbf{d}_t^2 \lesssim \int_0^t \exp\Big(-\int_s^t \frac{\bar{c}C_\eta \tilde{\beta}_0}{1 - e^{-2(T-r)}}\, \mathrm{d}r\Big) \Big((\mathbf{E}_t^{\text{strong}})^2 + \frac{\sqrt{(d + \mathbf{M}_2^2)T}}{\varepsilon_{\text{score}}}(\mathbf{E}_t^{\text{weak}})^2\Big)\, \mathrm{d}s.$$

If we substitute in the definitions of $\mathbf{E}_t^{\text{weak}}, \mathbf{E}_t^{\text{strong}}$,

$$\mathbf{d}_t^2 \lesssim \int_0^t \exp\Big(-\int_s^t \frac{\bar{c}C_\eta \tilde{\beta}_0}{1 - e^{-2(T-r)}}\, \mathrm{d}r\Big) \Big(\varepsilon_{\text{score}}^2 \frac{1 - e^{-2(T-s)}}{\tilde{\beta}_0} + \frac{\varepsilon_{\text{score}}^2\left(1 - e^{-2(T-s)}\right)^2}{\tilde{\beta}_0^2 (d + \mathbf{M}_2^2)T}\, \partial_s \mathbf{g}_s^2\Big)\, \mathrm{d}s$$

$$= \int_0^t \Big(\frac{e^{2(T-t)} - 1}{e^{2(T-s)} - 1}\Big)^{\frac{\bar{c}C_\eta \tilde{\beta}_0}{2}} \Big(\varepsilon_{\text{score}}^2 \frac{1 - e^{-2(T-s)}}{\tilde{\beta}_0} + \frac{\varepsilon_{\text{score}}^2\left(1 - e^{-2(T-s)}\right)^2}{\tilde{\beta}_0^2 (d + \mathbf{M}_2^2)T}\, \partial_s \mathbf{g}_s^2\Big)\, \mathrm{d}s.$$

Let us now simplify some of these integrals. First, for $K := \bar{c}C_\eta \tilde{\beta}_0 \gg 1$, and using the change of variables $v = e^{-2(T-s)}$, $\mathrm{d}v = 2v\, \mathrm{d}s$,

$$\int_0^t \Big(\frac{e^{2(T-t)} - 1}{e^{2(T-s)} - 1}\Big)^{\frac{\bar{c}C_\eta \tilde{\beta}_0}{2}} \left(1 - e^{-2(T-s)}\right)\, \mathrm{d}s = (e^{2(T-t)} - 1)^K \int_0^t (v^{-1} - 1)^{-K} v\, (v^{-1} - 1)\, \mathrm{d}s$$

$$= \frac{(e^{2(T-t)} - 1)^K}{2} \int_{e^{-2T}}^{e^{-2(T-t)}} \Big(\frac{v}{1 - v}\Big)^{K-1}\mathrm{d}v.$$

Next, let $\omega := e^{1/K}$.

$$\int_{e^{-2T}}^{e^{-2(T-t)}} \Big(\frac{v}{1-v}\Big)^{K-1}\mathrm{d}v = \sum_{j\geq 0}\int_{\omega^j \leq (1-v)/(1-e^{-2(T-t)})\leq \omega^{j+1}} \Big(\frac{v}{1-v}\Big)^{K-1}\mathrm{d}v$$

$$\leq \frac{1}{(1-e^{-2(T-t)})^{K-1}}\sum_{j\geq 0}\frac{1}{\omega^{(K-1)j}}\int_{\omega^j \leq (1-v)/(1-e^{-2(T-t)})\leq \omega^{j+1}} v^{K-1}\,\mathrm{d}v$$

$$\leq \frac{1}{(1-e^{-2(T-t)})^{K-1}}\sum_{j\geq 0}\frac{e^{-2(K-1)(T-t)}}{\omega^{(K-1)j}}\big(1-e^{-2(T-t)}\big)\,\omega^j\,(\omega-1)$$

$$\lesssim \frac{e^{-2(K-1)(T-t)}}{K\,(1-e^{-2(T-t)})^{K-1}}\sum_{j\geq 0}\frac{1-e^{-2(T-t)}}{\omega^{(K-2)j}} \lesssim \frac{e^{-2(K-1)(T-t)}\,(1-e^{-2(T-t)})}{K\,(1-e^{-2(T-t)})^{K-1}}\,.$$

On the other hand, a naïve bound is

$$\int_{e^{-2T}}^{e^{-2(T-t)}} \Big(\frac{v}{1-v}\Big)^{K-1}\mathrm{d}v \leq \frac{\int_{e^{-2T}}^{e^{-2(T-t)}} v^{K-1}\,\mathrm{d}v}{(1-e^{-2(T-t)})^{K-1}} \leq \frac{e^{-2K(T-t)}}{K\,(1-e^{-2(T-t)})^{K-1}}\,.$$

Using the naïve bound for $T-t \gtrsim 1$, and the refined bound for $T-t \lesssim 1$, we obtain

$$\int_0^t \Big(\frac{e^{2(T-t)}-1}{e^{2(T-s)}-1}\Big)^{\frac{\bar{c}C_\eta\tilde{\beta}_0}{2}}\big(1-e^{-2(T-s)}\big)\,\mathrm{d}s \lesssim \frac{(1-e^{-2(T-t)})^2}{\tilde{\beta}_0}\,.$$

On the other hand, integrating by parts, letting

$$f(s,t) := \Big(\frac{e^{2(T-t)}-1}{e^{2(T-s)}-1}\Big)^{\frac{\bar{c}C_\eta\tilde{\beta}_0}{2}}\big(1-e^{-2(T-s)}\big)^2 = e^{-4(T-s)}\frac{(e^{2(T-t)}-1)^K}{(e^{2(T-s)}-1)^{K-2}}$$

which is increasing in $s$,

$$\int_0^t \Big(\frac{e^{2(T-t)}-1}{e^{2(T-s)}-1}\Big)^{\frac{\bar{c}C_\eta\tilde{\beta}_0}{2}}\big(1-e^{-2(T-s)}\big)^2\,\partial_s \mathsf{g}_s^2\,\mathrm{d}s = f(t,t)\,\mathsf{g}_t^2 - f(0,t)\,\mathsf{g}_0^2 - \int_0^t \partial_s f(s,t)\,\mathsf{g}_s^2\,\mathrm{d}s$$

$$\leq f(t,t)\,\mathsf{g}_t^2\,.$$

Together with Lemma Lemma 4, it yields

$$\int_0^t \Big(\frac{e^{2(T-t)}-1}{e^{2(T-s)}-1}\Big)^{\frac{\bar{c}C_\eta\tilde{\beta}_0}{2}}\big(1-e^{-2(T-s)}\big)^2\,\partial_s \mathsf{g}_s^2\,\mathrm{d}s \lesssim d\,(1-e^{-2(T-t)}) + \mathtt{M}_2^2\,(1-e^{-2(T-t)})^2\,.$$

Finally, this all implies that

$$\mathsf{d}_t^2 \lesssim \frac{\varepsilon_{\text{score}}^2\,(1-e^{-2(T-t)})^2}{\tilde{\beta}_0^2} + \frac{\varepsilon_{\text{score}}^2}{\tilde{\beta}_0^2\,(d+\mathtt{M}_2^2)T}\big[d\,(1-e^{-2(T-t)}) + \mathtt{M}_2^2\,(1-e^{-2(T-t)})^2\big]\,.$$

Now, note that our bound on the KL divergence is given by

$$\mathbb{E}_{x\sim\pi_T}\,\mathsf{KL}(\mathbf{P}_x^{\mathsf{aux}}\,\|\,\mathbf{P}_x^{\mathsf{alg}}) \lesssim \int_0^{t_N} \eta_t^2 \mathsf{d}_t^2 \log\frac{1}{\mathsf{h}_t}\,\mathrm{d}t$$

$$\lesssim \big(\varepsilon_{\text{score}}^2 \log\frac{1}{h_N}\big)\int_0^{t_N}\Big(1 + \frac{d}{(d+\mathtt{M}_2^2)T\,(1-e^{-2(T-t)})}\Big)\,\mathrm{d}t$$

$$\lesssim \big(\varepsilon_{\text{score}}^2 \log\frac{1}{h_N}\big)\Big(T + \frac{1}{T}\log\frac{e^{2T}-1}{e^{2(T-t_N)}-1}\Big)\,.$$

$\square$

**Proof.** [Proof of Theorem Theorem 3] Lemma Lemma 11 states that

$$\mathsf{KL}(\pi_{t_N}^{\mathsf{aux}}\,\|\,\pi_{t_N}^{\mathsf{alg}}) \lesssim \big(\varepsilon_{\text{score}}^2 T + \frac{\varepsilon_{\text{score}}^2}{T}\log\frac{1}{T-t_N}\big)\log\frac{\tilde{\beta}_0\sqrt{(d+\mathtt{M}_2^2)T}}{\varepsilon_{\text{score}}\,(T-t_N)} + \mathsf{KL}(\pi_T\,\|\,\gamma)\,.$$

On the other hand, we have the following:

(1) For $T - t_N \lesssim 1$, we have
$$W_2^2(\pi_0, \pi_{T-t_N}) \lesssim \mathtt{M}_2^2 \, (T - t_N)^2 + d \, (T - t_N) \,.$$

(2) Via Lemma Lemma 11 again,
$$W_2^2(\pi_{t_N}^{\mathsf{aux}}, \pi_{T-t_N}) \lesssim \varepsilon_{\mathrm{score}}^2 \, (T - t_N)^2 + \frac{\varepsilon_{\mathrm{score}}^2 \, (T - t_N)}{T} \,.$$

(3) Lastly, since we use a Gaussian in place of $\pi_T$ as the initial distribution, we need to pay the additional factor
$$\mathsf{KL}(\pi_T \parallel \gamma) \leq e^{-T}(d + \mathtt{M}_2^2) \,,$$
using Chen et al. (2023a, Lemma 9). So we take $T \asymp \log \frac{d + \mathtt{M}_2^2}{\varepsilon_{\mathrm{score}}^2}$.

Thus, we should take $T \asymp \log \frac{d + \mathtt{M}_2^2}{\varepsilon_{\mathrm{score}}^2} \vee 1$, $T - t_N \asymp \frac{\varepsilon_{\mathrm{score}}^2}{d} + \frac{\varepsilon_{\mathrm{score}}}{\mathtt{M}_2}$. This all implies that
$$W_2^2(\pi_{t_N}^{\mathsf{aux}}, \pi_0) \lesssim \varepsilon_{\mathrm{score}}^2 \,, \qquad \mathsf{KL}(\pi_{t_N}^{\mathsf{aux}} \parallel \pi_{t_N}^{\mathsf{alg}}) = \widetilde{O}\big(\varepsilon_{\mathrm{score}}^2 \, (1 + \log^2\{\tilde{\beta}_0(d + \mathtt{M}_2^2)\})\big) \,.$$
From our choice of step sizes, we note that this takes $N$ steps with
$$N \asymp \frac{\tilde{\beta}_0 \sqrt{d + \mathtt{M}_2^2} \, T^{3/2}}{\varepsilon_{\mathrm{score}}} + \frac{\tilde{\beta}_0 \sqrt{(d + \mathtt{M}_2^2) T}}{\varepsilon_{\mathrm{score}}} \log \frac{1}{T - t_N} = \widetilde{\Theta}\Big(\frac{\tilde{\beta}_0 \sqrt{d + \mathtt{M}_2^2}}{\varepsilon_{\mathrm{score}}}\Big) \,.$$
$\square$

# B  EXAMPLES SATISFYING ASSUMPTION ASSUMPTION 3

We provide some examples of distributions where Assumption Assumption 3 holds for the true scores, i.e., for $\mathsf{s}_t = \nabla \log \pi_t$. The following examples all come from the literature on quantitative Lipschitz estimates of Kim–Milman maps (i.e., flow map for the probability flow ODE) which were originally used to establish log-Sobolev inequalities. For completeness, we provide derivations below.

- **Log-concave measures.** Let $\pi_0 \propto \exp(-V)$ with $\nabla^2 V \succeq 0$. Then, Assumption Assumption 3 holds with $\tilde{\beta}_0 \leq 1$.
- **Lipschitz perturbations of strongly log-concave measures.** Let $\pi_0 \propto \exp(-V - W)$, where $V$ is $\alpha$-strongly convex ($\alpha > 0$) and $W$ is $L$-Lipschitz. Then, Assumption Assumption 3 holds with $\tilde{\beta}_0 \leq L^2/\alpha \vee 1$.
- **Semi-log-concave over compact sets.** Let $\pi_0 \propto \exp(-V)$ over a compact set with diameter at most $R$, and such that $\nabla^2 V \succeq \alpha I_d$ for some $\alpha < 0$. Then, Assumption Assumption 3 holds with $\tilde{\beta}_0 \lesssim 1 \vee |\alpha| R^2$.
- **Gaussian convolutions of compactly supported measures.** Let $\pi_0 = \nu * \mathcal{N}(0, I_d)$, where $\nu$ has compact support, of diameter at most $R$. Then, Assumption Assumption 3 holds with $\tilde{\beta}_0 \lesssim 1 \vee R^2$.
- **Strongly log-concave outside a ball.** Let $\pi_0 \propto \exp(-V)$, where $V$ satisfies
$$\inf_{\|x-y\|=r} \frac{\langle \nabla V(x) - \nabla V(y), x - y \rangle}{\|x - y\|^2} \geq \begin{cases} \alpha - \beta, & \|x - y\| \leq R, \\ \alpha, & \|x - y\| > R, \end{cases}$$
for some $\alpha, \beta, R > 0$. Then, Assumption Assumption 3 holds with some constant $\tilde{\beta}_0$ depending only on $\alpha, \beta$, and $R$.

We remark that in all of these examples except the first, the log-Sobolev constant of $\pi_0$ scales exponentially in $\tilde{\beta}_0$, whereas our convergence bounds only scale polynomially in $\tilde{\beta}_0$. This implies that, given access to an accurate score estimator, diffusion models are far superior to standard MCMC methods such as the Langevin diffusion.

We also provide one instance showing the failure of Assumption Assumption 3.

- **Two point masses.** Consider $\pi_0 = \frac{1}{2} \delta_{\mathbf{e}_1} + \frac{1}{2} \delta_{-\mathbf{e}_1}$, where $\mathbf{e}_1$ is the vector $[1, 0, \ldots, 0]$. The Hessian is $-\nabla^2 \log \pi_t(\mathbf{x}) = \frac{1}{1 - e^{-2t}} I_d - \frac{e^{-2t}}{(1 - e^{-2t})^2} \mathbf{e}_1 \mathbf{e}_1^\top \operatorname{sech}^2\big(\frac{\operatorname{csch}(t)}{2} \langle \mathbf{e}_1, \mathbf{x} \rangle\big)$. Thus, along and near the critical strip $x_1 = 0$, the Hessian experiences blow-up at rate $1/t^2$ as $t \to 0$. This shows that there is no $\tilde{\beta}_0$ that suffices for all values of $\varepsilon_{\mathrm{score}}$.

This reasoning can be generalized to other mixtures of point masses.

## B.1 PROOFS

**Log-concave measures.** Let $\pi_0 \propto \exp(-V)$ where $V : \mathbb{R}^d \to \mathbb{R}$ is strongly log-concave. The conditional distribution of $X_t^{\to}$ given $X_0^{\to} = x_0$ is $N(e^{-t}x_0, (1 - e^{-2t})I_d)$. Using this, standard calculations give that

$$\nabla^2 \log \pi_t(x) = -\frac{I_d}{1 - e^{-2t}} + \frac{e^{-2t}}{(1 - e^{-2t})^2} \operatorname{cov}(X_0^{\to} \mid X_t^{\to} = x). \tag{B.1}$$

Now, the reverse conditional measure has the form

$$\pi_{0|t}(x \mid y) \propto \exp\Big(-\frac{\|y - e^{-t}x\|^2}{2(1 - e^{-2t})} - V(x)\Big),$$

so that

$$-\nabla^2 \log \pi_{0|t}(x \mid y) = \frac{e^{-2t}}{1 - e^{-2t}} I_d + \nabla^2 V(x) \succeq \frac{e^{-2t}}{1 - e^{-2t}} I_d. \tag{B.2}$$

The Brascamp–Lieb inequality (Brascamp & Lieb, 1976) then allows us to bound the covariance by the inverse of the matrix above. Thus, after some algebra,

$$\lambda_{\max}\Big(\nabla^2 \log \frac{\pi_t}{\gamma}\Big) = \lambda_{\max}(\nabla^2 \log \pi_t + I_d) \leq 1.$$

The minimum eigenvalue can be lower bounded in (B.1) by taking the covariance to be zero, which shows that $\tilde{\beta}_0 = 1$ is sufficient.

**Lipschitz perturbations of strongly log-concave measures.** Next, suppose $\pi_0 \propto \exp(-V - W)$, where $V$ is $\alpha$-strongly convex and $W$ is $L$-Lipschitz. The previous example showed that

$$\nabla^2 \log \frac{\pi_t}{\gamma} = \frac{1}{e^{2t} - 1} \Big(\frac{\operatorname{cov}_{\nu_{1-e^{-2t}, e^{-t}y}}}{1 - e^{-2t}} - I_d\Big),$$

where

$$\nu_{\tau,y}(\mathrm{d}x) \propto \exp\Big(-\frac{\|x - y\|^2}{2\tau} + \frac{\|x\|^2}{2}\Big) \pi(\mathrm{d}x).$$

Following the argument of Brigati & Pedrotti (2025),

$$\|\operatorname{cov}_{\nu_{\tau,y}}\|_{\mathrm{op}} \leq (\sqrt{\|\operatorname{cov}_{\tilde{\nu}_{\tau,y}}\|} + W_2(\nu_{\tau,y}, \tilde{\nu}_{\tau,y}))^2,$$

where

$$\tilde{\nu}_{\tau,y}(x) \propto \exp\Big(-\frac{\|x - y\|^2}{2\tau} + \frac{\|x\|^2}{2} - V(x)\Big).$$

Using Brascamp–Lieb, the first term is bounded by

$$\|\operatorname{cov}_{\tilde{\nu}_{\tau,y}}\| \leq \alpha - 1 + \frac{1}{\tau}.$$

On the other hand, by the $\mathsf{T}_2$ inequality and LSI,

$$W_2^2(\nu_{t,y}, \tilde{\nu}_{t,y}) \leq C_{\mathsf{LSI}}^2(\tilde{\nu}_{t,y}) \, \mathsf{FI}(\nu_{t,y} \parallel \tilde{\nu}_{t,y}).$$

The Fisher information is the expectation of the squared gradient norm of a $L$-Lipschitz function (namely $W$), whereas we use Bakry–Émery to bound the log-Sobolev constant. This all yields the bound on the covariance, for $\tau = 1 - e^{-2t}$:

$$\|\operatorname{cov}_{\nu_{\tau,y}}\| \leq \Big(\sqrt{\frac{1}{\alpha - 1 + \frac{1}{\tau}}} + \frac{L}{\alpha - 1 + \frac{1}{\tau}}\Big)^2$$

$$= \Big(\sqrt{\frac{1}{\alpha + e^{-2t}/(1 - e^{-2t})}} + \frac{L}{\alpha + e^{-2t}/(1 - e^{-2t})}\Big)^2$$

$$\leq \Big(\sqrt{\frac{1 - e^{-2t}}{e^{-2t}}} + \frac{L}{2\sqrt{\alpha e^{-2t}/(1 - e^{-2t})}}\Big)^2 \lesssim \big(1 \vee \frac{L^2}{\alpha}\big) \frac{1 - e^{-2t}}{e^{-2t}}.$$

In particular, this implies the existence of an estimator in Assumption Assumption 3 with $\tilde{\beta}_0 \lesssim 1 \vee L^2/\alpha$.

**Semi-log-concave measures over compact sets.** Let $\pi_0 \propto \exp(-V)$ over a compact set with diameter at most $R$, and such that $\nabla^2 V \succeq \alpha I_d$ for some $\alpha < 0$. By (B.1) and (B.2), when $e^{-2t}/(1-e^{-2t}) \geq -2\alpha$, then $\lambda_{\max}(\nabla^2 \log(\pi_t/\gamma)) \lesssim 1$. On the other hand, when $e^{-2t}/(1-e^{-2t}) \leq -2\alpha$, then

$$\lambda_{\max}\left(\nabla^2 \log \frac{\pi_t}{\gamma}\right) \leq \frac{e^{-2t}}{(1-e^{-2t})^2}\, R^2 \leq \frac{-2\alpha R^2}{1-e^{-2t}}\,.$$

Putting together the two cases, $\tilde{\beta}_0 \lesssim 1 \vee |\alpha| R^2$. This example and the next are taken from Mikulincer & Shenfeld (2023; 2024).

**Gaussian convolutions of compactly supported measures.** Let $\pi_0 = \nu * \mathcal{N}(0, I_d)$, where $\nu$ has compact support, of diameter at most $R$. A similar computation to the above examples readily yields

$$\nabla^2 \log \frac{\pi_t}{\gamma} \preceq R^2 e^{-2t}\, I_d\,.$$

Therefore, we can take $\tilde{\beta}_0 \lesssim 1 \vee R^2$.

**Strongly log-concave outside a ball.** This example is taken from Conforti et al. (2025b). The constant was not explicitly computed therein in terms of $\alpha$, $\beta$, and $R$.

## C EXPERIMENTAL DETAILS

### C.1 ADAPTING THE OU PROCESS TO THE EDM FRAMEWORK

Clearly (rev-OU) fits the general SDE (SDE) by taking $\lambda(t) = -1$, $f_t(X_t) = 2\, \nabla \log(\pi_{T-t}/\gamma)(X_t)$, and $g(t) = \sqrt{2}$. We wish to write (rev-OU) in terms of (EDM). The EDM forward process is defined as $X_t = c(t)\, X_0 + c(t)\, \sigma(t)\, z$ while (OU) admits the closed-form solution $X_t = e^{-t} X_0 + B_{1-e^{-2t}}$ where $B_{\cdot}$ denotes the Wiener process. By comparison, we read $c(t) = e^{-t}$, $\sigma(t) = \sqrt{e^{2t}-1}$. Alternatively, we realize that the OU process is a special case of the VP SDE (Song et al., 2021b) when $\beta_{\min} = \beta_{\max} = 2$, ($\beta_d := \beta_{\max} - \beta_{\min} = 0$) and read from Table 1 of Karras et al. (2022). Matching $\sqrt{2\beta(t)}\, \sigma(t)\, c(t)$ to $\sqrt{2}$, we find that $\beta(t) = (\sigma(t)\, c(t))^{-2}$. Using the relationship between the forward and reverse SDE (Karras et al., 2022, eq. (6)), we have recovered (OU) and (rev-OU).

It is helpful to remember that the score $\hat{s}_t(X_t)$ is internally implemented with *denoising score matching* (Hyvärinen, 2005; Vincent, 2011) and admits the formula

$$\hat{s}_t(x) = \frac{D(x/c(t); \sigma(t)) - x/c(t)}{c(t)\, \sigma(t)^2}\,, \tag{EDM-score}$$

where $D(\cdot; \sigma)$ is a neural network denoiser trained to predict the unnoised $x$ given $x + \sigma z$, $z \sim \gamma$. Writing (EDM) in terms of the score instead of the denoiser allows for a cleaner implementation which is closer to the SDE, especially to implement our suggestions around the time scaling $\lambda(t)$.

### C.2 VARIANTS OF THE RANDOMIZED MIDPOINT

Our starting point is the semi-linear SDE (SDE)

$$dX_t = (\lambda(t) X_t + f_t(X_t))\, dt + g(t)\, dB_t\,. \tag{C.1}$$

From the intuition that a linear SDE of the form $dX_t = \lambda(t) X_t\, dt + g(t)\, dB_t$ admits a closed-form solution, we use the ODE integrating factor $\omega(t) := \exp(-\int_{t_0}^t \lambda)$ as an ansatz. By Itô's rule,

$$\begin{aligned}
d(\omega(t) X_t) &= (d\omega(t))\, X_t + \omega(t)\, dX_t \\
&= -\lambda(t)\, \omega(t) X_t\, dt + \omega(t)\, [(\lambda(t) X_t + f_t(X_t))\, dt + g(t)\, dB_t] \\
&= \omega(t)\, f_t(X_t)\, dt + \omega(t)\, g(t)\, dB_t\,,
\end{aligned}$$

where we have successfully removed the linear term. Integrating both sides from some starting time $t_0$ to $t_0 + h$, we have the integral representation

$$\omega(t_0 + h)X_{t_0+h} - \omega(t_0)X_{t_0} = \int_{t_0}^{t_0+h} \omega(t) f_t(X_t) \, dt + \int_{t_0}^{t_0+h} \omega(t) g(X_t) \, dB_t \,,$$

$$X_{t_0+h} = \frac{\omega(t_0)}{\omega(t_0 + h)} X_{t_0} + \int_{t_0}^{t_0+h} \frac{\omega(t)}{\omega(t_0 + h)} f_t(X_t) \, dt + \int_{t_0}^{t_0+h} \frac{\omega(t)}{\omega(t_0 + h)} g(X_t) \, dB_t \,. \quad \text{(INT)}$$

In order to approximate (INT) we perform a two-step discretization scheme. First, we draw a random time $\tau$ from the density proportional to $t \mapsto \mathbb{1}_{[t_0,t_0+h]}(t)\,\omega(t)$ which serves as our midpoint. Defining $\Omega(t) := \int_{t_0}^t \omega$, explicitly

$$\tau \sim p(\tau) = \begin{cases} \frac{\omega(t)}{\Omega(t_0+h)-\Omega(t_0)}, & t_0 \leq \tau \leq t_0 + h \,; \\ 0, & \text{otherwise}\,. \end{cases}$$

We can use the plug-in estimator $(\Omega(t_0 + h) - \Omega(t_0))f_{t_0+\tau}(X_{t_0+\tau})/\omega(t_0 + h)$ to obtain an unbiased estimate to the integral in the fashion of Monte Carlo quadrature. Unfortunately we do not know the value of $X_{t_0+\tau}$, necessitating a second approximation. We use an Euler scheme, assuming that the function is constant on the interval and taking the left endpoint (which we do know). Thus, we have

$$X^+_{t_0+\tau} = \frac{\omega(t_0)}{\omega(t_0 + \tau)} X_{t_0} + \frac{\Omega(t_0 + \tau)}{\omega(t_0 + \tau)} f_{t_0}(X_{t_0}) + \int_{t_0}^{t_0+\tau} \frac{\omega(t)}{\omega(t_0 + \tau)} g(X_t) \, dB_t \,,$$

$$X_{t_0+h} = \frac{\omega(t_0)}{\omega(t_0 + h)} X_{t_0} + \frac{\Omega(t_0 + h)}{\omega(t_0 + h)} f_{t_0+\tau}(X_{t_0+\tau}) + \int_{t_0}^{t_0+h} \frac{\omega(t)}{\omega(t_0 + h)} g(X_t) \, dB_t \,.$$

It remains to treat the noise terms. Define the stochastic process

$$Y_t := \int_{t_0}^t \frac{\omega(t')}{\omega(t_0 + h)} g(t') \, dB_{t'} \,.$$

Clearly $\mathbb{E}[Y_t] = 0$ and

$$\text{Var}[Y_t] = \mathbb{E}[Y_t^2] = \int_{t_0}^t \frac{\omega(t')^2}{\omega(t_0 + h)^2} g(t')^2 \, dt' \,,$$

by an application of Itô's rule. We will compute $(\xi^+, \xi) \sim (Y_{t_0+\tau}, Y_{t_0+h})$ by conditional simulation. Defining $\eta(t) := \int_{t_0}^t (\omega g)^2$; if $z^+ \sim \gamma$ then $\xi^+ = [\sqrt{\eta(t_0 + \tau) - \eta(t_0)}/\omega(t_0 + \tau)] z^+$ has the right marginal distribution. Next, we perform the domain decomposition

$$Y_{t_0+h} = \int_{t_0}^{t_0+\tau} \frac{\omega(t)}{\omega(t_0 + h)} g(t) \, dB_t + \int_{t_0+\tau}^{t_0+h} \frac{\omega(t)}{\omega(t_0 + h)} g(t) \, dB_t$$

$$= \frac{\omega(t_0 + \tau)}{\omega(t_0 + h)} \int_{t_0}^{t_0+\tau} \frac{\omega(t)}{\omega(t_0 + \tau)} g(t) \, dB_t + \int_{t_0+\tau}^{t_0+h} \frac{\omega(t)}{\omega(t_0 + h)} g(t) \, dB_t$$

$$= \frac{\omega(t_0 + \tau)}{\omega(t_0 + h)} Y_{t_0+\tau} + \int_{t_0+\tau}^{t_0+h} \frac{\omega(t)}{\omega(t_0 + h)} g(t) \, dB_t \,,$$

and make use of the fact that the latter term is independent of $Y_{t_0+\tau}$ and normally distributed with mean 0 and variance $(\eta(t_0 + h) - \eta(t_0 + \tau))/\omega(t_0 + h)^2$. Thus, we can compute for $z \sim \gamma$ independent of $z^+$, $\xi = [\omega(t_0 + \tau)/\omega(t_0 + h)] \xi^+ + [\sqrt{(\eta(t_0 + h) - \eta(t_0 + \tau))}/\omega(t_0 + h)] z$.

Putting everything together, we have the following generalization of Algorithm 1.

---

**Algorithm 2:** Generalized randomized midpoint kernel on $[t_0, t_1]$

---

**Input:** current state $X_{t_0} \in \mathbb{R}^d$; step $h := t_1 - t_0$; drift $f_t(\cdot)$; noise term $g(\cdot)$.

**Input:** scaling factor $\lambda(t)$; integrating factor $\omega(t) := \exp(-\int_{t_0}^t \lambda)$; normalizing factor
$\Omega(t) := \int_{t_0}^t \omega$; inverse $\Omega^{-1}(\cdot)$; noise factor $\eta(t) := \int_{t_0}^t (\omega g)^2$.

**1. Draw the randomized midpoint.** Sample $U \sim \mathsf{Unif}(0, 1)$ and set

$$\tau = \Omega^{-1}((1 - U)\,\Omega(t_0) + U\,\Omega(t_1)) \quad \text{i.e., with density } p(\tau) \propto 1_{[t_0, t_1]}(t)\,\omega(t).$$

**2. Midpoint prediction for $X_{t_0+\tau}^+$.** Draw $Z_1 \sim \mathcal{N}(0, I_d)$ and set the OU noise
$\xi^+ := [\sqrt{|\eta(t_0 + \tau)|}/\omega(t_0 + \tau)]\,Z_1$. Then

$$X_{t_0+\tau_k}^+ = \frac{\omega(t_0)}{\omega(t_0 + \tau)}\,X_{t_0} + \frac{\Omega(t_0 + \tau)}{\omega(t_0 + \tau)}\,f_{t_0}(X_{t_0}) + \xi^+\,.$$

**3. Full-step update for $X_{t_1}$.** Draw $Z_2 \sim \mathcal{N}(0, I_d)$ independent of $Z_1$ and set

$$\xi = \frac{\omega(t_0 + \tau)}{\omega(t_1)}\,\xi^+ + \frac{\sqrt{|\eta(t_1) - \eta(t_0 + \tau)|}}{\omega(t_1)}\,Z_2\,.$$

Compute the score at the randomized time and update

$$X_{t_1} = \frac{\omega(t_0)}{\omega(t_1)}\,X_{t_0} + \frac{\Omega(t_1)}{\omega(t_1)}\,f_{t_0+\tau}(X_{t_0+\tau}^+) + \xi\,.$$

---

Note that we take the absolute value in the computation of $(\xi^+, \xi)$ so that Algorithm 2 is valid also in reverse time (i.e., when $t_1 < t_0$ and $h < 0$).

For example, one concrete instantiation as mentioned in the main text is given by

$$\begin{aligned}
X_{t_{k-1}+\tau_k}^+ &= X_{t_{k-1}} + \tau_k f_{t_{k-1}}(X_{t_{k-1}}) + \text{noise}\,, \\
X_{t_k} &= X_{t_{k-1}} + h_k f_{t_{k-1}+\tau_k}(X_{t_{k-1}+\tau_k}) + \text{noise}\,,
\end{aligned} \tag{RME}$$

which corresponds to randomized midpoint without exponential Euler when $\lambda(t) = 0$.

### C.2.1 IMPLEMENTATION DETAILS

The quantity $\omega(t)$ is free up to multiplicative factor and $\Omega(t), \eta(t)$ are free up to constants, assuming they agree with each other. It sometimes convenient to arbitrarily base the integrals at $t_0$, i.e. to compute $\Omega(t) = \int_{t_0}^t \omega(t)\,dt$, resulting in definite integrals for the differences in integrated quantities in Algorithm 2. When it is not possible to analytically integrate $\omega, \Omega$, or $\eta$ or to invert $\Omega$, numerical quadrature and root finding can be used instead. We use `scipy.integrate.quad` and `scipy.optimize.root_scalar` respectively for these tasks, from the SciPy library (Virtanen et al., 2020). For quadrature it can help to signal discontinuities like $S_{\text{tmin}}$ and $S_{\text{tmax}}$ with the `points` argument. For root finding we use the `"brentq"` method with interval $[t_0, t_1]$. Although in principle we could use a higher-order method like the `"halley"` method since $\Omega$ is twice differentiable with derivatives $\Omega' = \omega, \Omega''(t) = -\lambda(t)\,\omega(t)$, in our settings we find both quadrature and root finding to converge to near machine precision ($\approx 10^{-11}$–$10^{-15}$) in a handful of iterations ($< 10$).

### C.2.2 CONCRETE CHOICES OF SCALING FACTOR

Following (EDM), we see that in order for the drift to be a time-scaling of the score, it suffices to take $\lambda(t) = \dot{c}(t)/c(t)$. For the drift to be a time-scaling of the relative score, we take

$$\lambda(t) = \frac{\dot{c}(t)}{c(t)} + \frac{c(t)^2 \dot{\sigma}(t)\sigma(t)}{\sigma_T^2} + \frac{c(t)^2 \beta(t)\sigma(t)^2}{\sigma_T^2}\,,$$

where $\sigma_T^2$ is the variance of the forward process at time $T$, $\pi_T$. We also consider a "network-adapted" strategy (as opposed to the aforementioned "SDE-adapted") strategy by expanding the score in terms

of the denoiser (EDM-score) and collecting linear terms, resulting in $\lambda(t) = \dot{c}(t)/c(t) + \dot{\sigma}(t)/\sigma(t)$. We can also account for the skip connection in the denoiser itself, resulting in the choice of

$$\lambda(t) = \frac{\dot{c}(t)}{c(t)} + \left(1 - c_{\text{skip}}(t)\right) \frac{\dot{\sigma}(t)}{\sigma(t)},$$

where $c_{\text{skip}}(t)$ is the skip connection in the denoiser $D(\cdot; \sigma)$ (see Table 1 of Karras et al. (2022)). In particular, we consider $c_{\text{skip}}(t) = \sigma_{\text{data}}^2/(\sigma(t)^2 + \sigma_{\text{data}}^2)$ for $\sigma_{\text{data}} = 0.5$.

In our experiments we use the relative score for the OU and VP processes, the non-relative score for the EDM process, and the skip connection for the VE process.

### C.3 STOCHASTIC SAMPLING

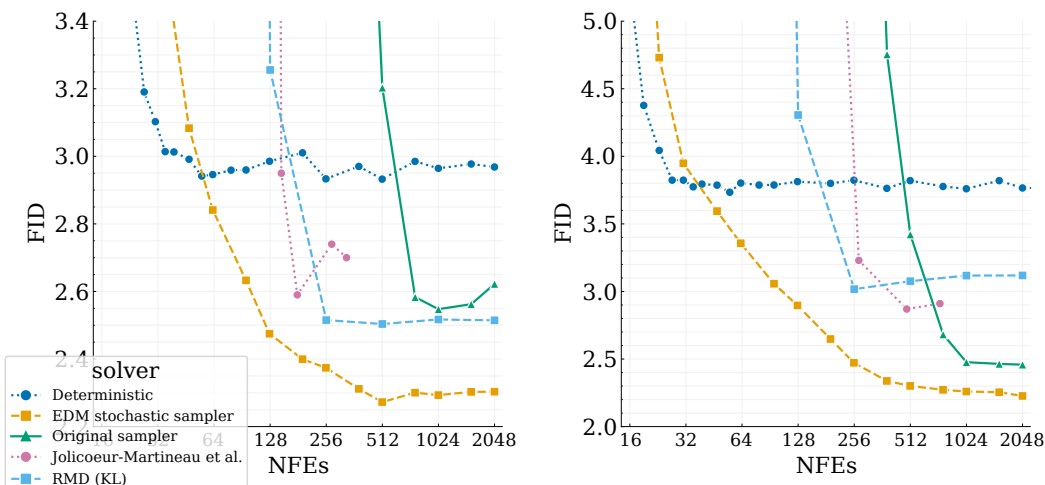

Figure 4: **Quantitative results: SDE sampling.** Image quality as measured by FID↓ with number of score function evaluations (NFEs) for unconditional CIFAR-10 with the VE (left panel) and VP (right panel) models of Song et al. (2021b). "Deterministic" is the Heun ODE solver proposed by Karras et al. (2022) (Algorithm 1). "EDM stochastic sampler" is the stochastic solver of Karras et al. (2022) (Algorithm 2). "Original sampler" is Euler-Maruyama for VP and predictor-corrector for VE, following Song et al. (2021b). "Jolicoeur-Martineau et al." is the solver of Jolicoeur-Martineau et al. (2021). Note that all data except for (RMD) is taken verbatim from Karras et al. (2022). In particular, they take the minimum of three FID computations, which we do not do for (RMD).

We test stochastic sampling in the $32 \times 32$ unconditional CIFAR-10 dataset (Krizhevsky, 2009) with both the VP and VE models of Song et al. (2021a), following the experiments of Karras et al. (2022). Our results are shown in Figure 4. Although (RMD) is not state of the art, it achieves a competitive FID, often in much fewer steps than comparable solvers (e.g. Euler-Maruyama and predictor-corrector of Song et al. (2021b) and the second order solver of Jolicoeur-Martineau et al. (2021)).

All solvers seem to asymptote to a fixed FID with enough sampling steps; we find in preliminary numerical experiments that the choice of noise coefficient $\beta(t)$ has a significant effect on the this FID. We use the noise coefficient $\beta(t) = \dot{\sigma}(t)/\sigma(t)$ that minimizes an upper bound on the KL divergence (Ma et al. (2024), Section 2.4). This is also the implicit choice of Song et al. (2021a) and under this choice the forward process of (EDM) does not contain the score, as noted by Karras et al. (2022). One key advantage of this choice is that it is hyperparameter free, in contrast to the stochastic sampler of Karras et al. (2022) which involves four hyperparameters all tuned with an expensive grid search. For (RMD) we use these tuned parameters (Appendix E.2 "Stochastic sampling parameters" of Karras et al. (2022)) verbatim. Indeed, we find that the tuned parameters systematically improve FID while simultaneously *degrading* $\text{FD}_{\text{DINOv2}}$, suggesting overfitting to FID. We defer a detailed investigation of the choice of noise coefficient and a "clean room" $\text{FD}_{\text{DINOv2}}$ comparision to future work.

## C.4 ADDITIONAL FIGURES

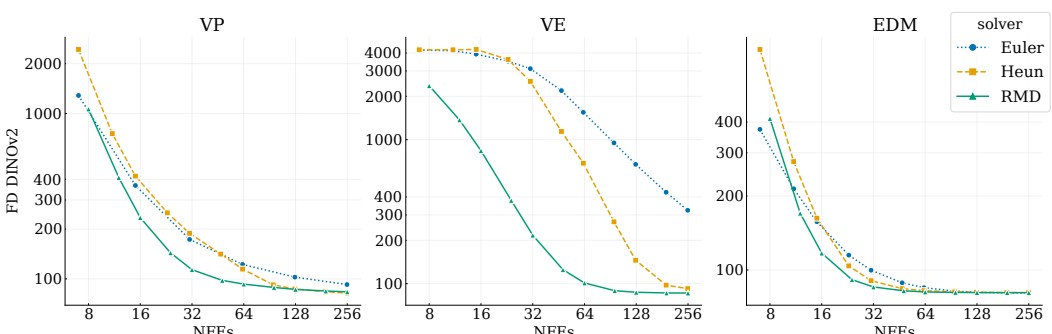

Figure 5: Image quality as measured by FDDINOv2. Figure 3 uses the same generated images.

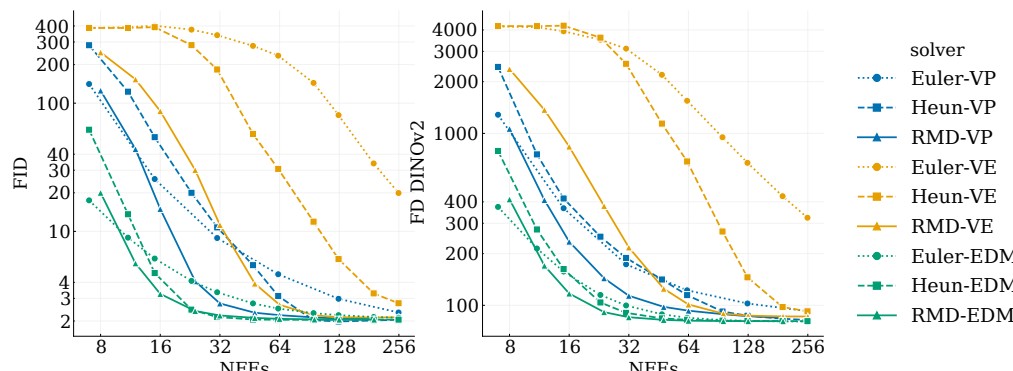

Figure 6: A variant of Figure 3 with all methods and settings shown on the same scale.