# OpenReview forum: "Sublinear iterations can suffice even for DDPMs"
_ICLR.cc/2026/Conference — Submitted to ICLR 2026_

### Official Review · Reviewer_qw3a · 2025-10-29

**Soundness:** 2
**Presentation:** 2
**Contribution:** 3
**Rating:** 4
**Confidence:** 3

**Summary:**

The paper present an analysis of a stochastic integrator for the DDPM SDE.
It is based on an intermediate random middle point that is used to estimate evaluate the score in place of the initial point of the time interval.
A sublinear convergence theorem is proven: the bound is in KL divergence towards an intermediary distribution that is close to the data distribution in Wasserstein distance.
Numerical experiments tend to support the superiority of the proposed sampler.

**Strengths:**

The proposed sampler is sound and rely on previous literature.
Theorem 3 provides sublinear complexity bound (towards a somewhat obscure intermediate probability).
The appendix material for the proof of Theorem 3 seems well-written and documented (did not check the proof).

**Weaknesses:**

From the abstract one can read "prior works which obtained such bounds worked instead with ODE-based sampling and had to make modifications to the sampler which deviate from how they are used in practice."
A similar claim line 304 "This is an option that we cannot afford in this work, as our goal is to simply analyze a discretization of the vanilla DDPM reverse process without further algorithmic modifications."
But the proposed work studies Algorithm 1 that:
* requires two score evaluation per iteration (OK but should be hilighted)
* is proven convergent using some specific decaying step size only discussed in Appendix (Equation A.2 line 954)
* In addition, the convergence is only proven through the use of an intermediate distribution $\pi^{\mathrm{approx}}$, with a mixed role for KL divergence and $W_2$-distance (see discussion line 284).

Due to the difference in sampling schemes, the comparison experiments in Section 5 lack clarity.
For the OU process, what are the step size used for EMD and EED?
For RMD, is it the step size from Equation (A.2)?
Why isn't this choice discussed in the main paper?
Why figure 1 stops at 64 NFEs while standard DDPM would use 1k or 2k steps (Ho et al 2020, Song et al 2021)?
Why is there no comparison with a predictor-corrector scheme that is computationally closer (two score evaluations and two Gaussian noise per iteration)?

"Figure 3: Quantitative results: Deterministic sampling:" Is RMD deterministic?

Minor remarks:
* the presentation of Equation RMD could be made more consistent with Algorithm 1 by using $t_{k-1}+\tau_k$ instead of $t$
* DDRaM not defined in the main text (but in abstract)
* Figure 4 and 5 are in the appendix, which is not clear when reading the text line 463 and 470.
* line 744: $Y^{\mathrm{aux}}$ or $Y^{\mathrm{alg}}$ ?
* line 755: Thenn

**Questions:**

See questions in weaknesses.

---

> ### Author Response · Authors · 2025-11-19
> **Response to Reviewer qw3a**
>
> The appendix material for the proof of Theorem 3 seems well-written and documented (did not check the proof).
>
> - Thank you. In fact, we have edited the appendix further to make the proof easier to follow (in particular, we have made the integral computations simpler and more explicit in the proof of Theorem 3).
>
> From the abstract one can read "prior works which obtained such bounds worked instead with ODE-based sampling and had to make modifications to the sampler which deviate from how they are used in practice." A similar claim line 304 "This is an option that we cannot afford in this work, as our goal is to simply analyze a discretization of the vanilla DDPM reverse process without further algorithmic modifications." But the proposed work studies Algorithm 1 that: requires two score evaluation per iteration (OK but should be hilighted), is proven convergent using some specific decaying step size only discussed in Appendix (Equation A.2 line 954). In addition, the convergence is only proven through the use of an intermediate distribution with a mixed role for KL divergence and $W_2$-distance (see discussion line 284).
>
> - We concede our approach is more complicated than Euler DDPM, but significantly simpler than the approach in some of the comparable theoretical work, e.g. [1]. We highlight that the requirement of two score evaluations per iteration is not at all unreasonable, as practitioners use the Heun sampler, which also requires two evaluations per iteration [2]. [2] also proposes a step size schedule that decays the step size when the sample is further into the sampling process (section D of the appendix). By “further algorithmic modifications” we mean schemes like predictor-corrector as employed in previous theoretical works like Chen et al. (2023c). In contrast, we wish to analyze an algorithm that is simple and easy to implement, which is as close to a discretization of a SDE as possible.
>
> [1] G. Li, and Y. Jiao. "Improved convergence rate for diffusion probabilistic models." The Thirteenth International Conference on Learning Representations. 2024.
>
> [2] T. Karras, M. Aittala, T. Aila, and S. Laine, “Elucidating the Design Space of Diffusion-Based Generative Models,” Oct. 11, 2022, arXiv: arXiv:2206.00364. doi: 10.48550/arXiv.2206.00364.

---

> ### Author Response · Authors · 2025-11-19
> **Response to Reviewer qw3a, continued**
>
> Due to the difference in sampling schemes, the comparison experiments in Section 5 lack clarity. For the OU process, what are the step size used for EMD and EED? For RMD, is it the step size from Equation (A.2)? Why isn't this choice discussed in the main paper? Why figure 1 stops at 64 NFEs while standard DDPM would use 1k or 2k steps (Ho et al 2020, Song et al 2021)? Why is there no comparison with a predictor-corrector scheme that is computationally closer (two score evaluations and two Gaussian noise per iteration)?
>
> - The step sizes we use are the adjacent differences in a sequence of time steps. Since we are ablating over the solver, we keep the time steps fixed for all three solvers (EMD, EED, RMD) and use evenly spaced time steps according to the “VP” scheme [1] in Table 1 of [2] as a simple baseline. Note that although the time steps are associated with the “VP” column, they can be used with any choice of schedule and scaling because they ultimately determine the noise levels to evaluate at. We thank the reviewer for the attention to detail and we have added these experimental details to the revised draft.
>
> - Although the original DDPM [1] and score-based diffusion papers [2] used thousands of sampling steps, recent training and architectural improvements mean we can achieve good accuracy with much fewer steps (see Figure 2 of [3], for example). For completeness, we ran the samplers to 256 NFEs and this extended version of Figure 1 was added to the revised draft.
>
> - We plan on adding a more detailed comparison of various stochastic sampling algorithms beyond the OU process setting in the future. For now, we note that Figure 4 of [3] compares to the predictor-corrector scheme of [2], along with Euler-Maruyma and a higher-order adaptive SDE solver [4, 5] that also uses two score evaluations per step.
>
> [1] J. Ho, A. Jain, and P. Abbeel, “Denoising Diffusion Probabilistic Models,” Dec. 16, 2020, arXiv: arXiv:2006.11239. doi: 10.48550/arXiv.2006.11239.
>
> [2] Y. Song, J. Sohl-Dickstein, D. P. Kingma, A. Kumar, S. Ermon, and B. Poole, “Score-Based Generative Modeling through Stochastic Differential Equations,” Feb. 10, 2021, arXiv: arXiv:2011.13456. doi: 10.48550/arXiv.2011.13456.
>
> [3] T. Karras, M. Aittala, T. Aila, and S. Laine, “Elucidating the Design Space of Diffusion-Based Generative Models,” Oct. 11, 2022, arXiv: arXiv:2206.00364. doi: 10.48550/arXiv.2206.00364.
>
> [4] A. Jolicoeur-Martineau, K. Li, R. Piché-Taillefer, T. Kachman, and I. Mitliagkas, “Gotta Go Fast When Generating Data with Score-Based Models,” May 28, 2021, arXiv: arXiv:2105.14080. doi: 10.48550/arXiv.2105.14080.
>
> [5] A. J. Roberts, “Modify the Improved Euler scheme to integrate stochastic differential equations,” Oct. 02, 2012, arXiv: arXiv:1210.0933. doi: 10.48550/arXiv.1210.0933.
>
> "Figure 3: Quantitative results: Deterministic sampling:" Is RMD deterministic?
>
> - We thank the reviewer for pointing this out. We intended the figure caption to contrast “deterministic sampling” (i.e., with an ODE) to “stochastic sampling” (i.e., with a SDE). We realize this choice of terminology is ambiguous in the presence of a randomized ODE solver (RMD) that is nondeterministic, even in the ODE case. In our revised draft, we have changed the caption to “Quantitative results: ODE sampling”.

---

> ### Comment · Reviewer_qw3a · 2025-11-19
>
> I thank the authors for their answers to my comments.
> Can you please clarify one technical point: The algorithm is proven convergent using some specific decaying step size discussed in Appendix (Equation A.2 line 954). Is this the step size scheme used in the numerical experiments?

---

> > ### Author Response · Authors · 2025-11-20
> >
> > Thanks again for the follow-up question. In the experiments, we use a step size schedule that is different from the theoretical results, but which maps on to schemes that are used empirically, as originally introduced in the EDM paper [1]. While this means the experiments differ from the theory, we made this choice to facilitate a comparison between the relevant baselines and the existing literature, which would be challenging if we adopted the exact step size scheme proposed in our proof. On a technical level, exactly employing the step size schedule proposed in our theory would be difficult, because it depends on the Lipschitz constant of the neural network through $\tilde{\beta}_0$, which is unknown and challenging to estimate in practice.
> >
> > **Further Detail.** As mentioned in our previous reply, in Figures 1 & 2, we use the VP step size schedule from Table 1 of [1]. In Figure 3, we use the step size schedule corresponding to the choice of process again dictated by Table 1 of [1]. i.e., in each case we select the time steps, the schedule, and the scaling as recommended in the table. We emphasize that this corresponds to the original choices made in [2] for the VP and VE processes. We will add these details to the updated manuscript.
> >
> > Please let us know if there is any further clarification we can provide on the experiments or theory.
> >
> > **References.**
> >
> > [1] T. Karras, M. Aittala, T. Aila, and S. Laine, “Elucidating the Design Space of Diffusion-Based Generative Models,” Oct. 11, 2022, arXiv: arXiv:2206.00364. doi: 10.48550/arXiv.2206.00364.
> >
> > [2] Y. Song, J. Sohl-Dickstein, D. P. Kingma, A. Kumar, S. Ermon, and B. Poole, “Score-Based Generative Modeling through Stochastic Differential Equations,” Feb. 10, 2021, arXiv: arXiv:2011.13456. doi: 10.48550/arXiv.2011.13456.

---

### Official Review · Reviewer_wzAK · 2025-10-29

**Soundness:** 3
**Presentation:** 3
**Contribution:** 1
**Rating:** 4
**Confidence:** 4

**Summary:**

This work introduces the Denoising Diffusion Randomized Midpoint (DDRaM) method, a new SDE-based integrator for diffusion models that achieves sublinear $\sqrt{d}$ computational complexity in score evaluations under smoothness assumptions. Using the shifted composition rule, the authors provide the first theoretical sublinear convergence guarantee for pure DDPM sampling.

**Strengths:**

(1) The paper analyzes a stochastic sampler with the random point method, and proves that the KL and $W_2$ divergence can be controlled with iteration complexity that has a sublinear dependence on the dimension $d$.

(2) It conducts experiments to demonstrate the superiority of the randomized point method.

**Weaknesses:**

(1) The equations for reverse SDE seem incorrect. It can be seen from related works, e.g., [1], that there is no $\gamma$ there.

(2) The keyword, DDPM, is not very accurate. Actually, it should be termed as sampling diffusion with stochastic samplers, or SDE. This is because DDPM is only a special case of score-based diffusion when taking the limit in the length of time steps. However, the denoising diffusion model introduced in this paper is more related to the score-based SDE, instead of the original DDPM paper.

(3) The argument regarding why we approximate some distribution $\pi_{approx}$ is not convincing. The readers compare it with $\pi_{\delta}$ with early stopping. However, this is because when $\delta$ is small, $\pi_{\delta}$ is at least close to $\pi_0$ in $W_2$ distance. To make your argument reasonable, at least some similar $W_2$ guarantee should be provided. In contrast, in the main text, how it is defined is never specified. When I check the proof (Lemma 6), I find that this distribution even depends on the transition kernel of $P_{alg}$. As a result, the kind of guarantee is very unusual. And the comparison with prior works under this metric is unfair.

(4) The paper misses a closely related work [2] when introducing the analysis of DDIM.

---

[1] Nearly d-linear Convergence Bounds For Diffusion Models Via Stochastic Localization Benton et al., 2024 ICLR

[2] Unified Convergence Analysis for Score-Based Diffusion Models with Deterministic Samplers ICLR

**Questions:**

See weaknesses

---

> ### Author Response · Authors · 2025-11-19
> **Response to Reviewer wzAK**
>
> The equations for reverse SDE seem incorrect. It can be seen from related works, e.g., [1], that there is no $\upgamma$ there.
>
> - This concern seems to stem from a misunderstanding: our equations are a valid discretization of the reverse SDE and are inspired by the formulation in Conforti et al. [1]. The addition of $\upgamma$ to the score is compensated by taking the linear term to be $-X_t$ instead of $X_t$ in the drift. This is helpful as the relative score satisfies some nice properties which permit more elegant bounds.
>
> [1] Conforti, Giovanni, Alain Durmus, and Marta Gentiloni Silveri. "KL convergence guarantees for score diffusion models under minimal data assumptions." SIAM Journal on Mathematics of Data Science 7.1 (2025): 86-109.
>
> The keyword, DDPM, is not very accurate. Actually, it should be termed as sampling diffusion with stochastic samplers, or SDE. This is because DDPM is only a special case of score-based diffusion when taking the limit in the length of time steps. However, the denoising diffusion model introduced in this paper is more related to the score-based SDE, instead of the original DDPM paper.
>
> - We acknowledge that "DDPM", as introduced in its original sense [1], refers to a more specific discretization scheme than that intended in our work. However, we use the terminology to refer generally to score-based SDE sampling, as is common in the literature. See [2] for some examples where "DDPM" has been conflated with score-based SDEs in the literature. We will include a footnote expounding this point in a revised draft.
>
> [1] J. Ho, A. Jain, and P. Abbeel, “Denoising Diffusion Probabilistic Models,” Dec. 16, 2020, arXiv: arXiv:2006.11239. doi: 10.48550/arXiv.2006.11239.
>
> [2] Nakkiran, Preetum, et al. "Step-by-step diffusion: An elementary tutorial." arXiv preprint arXiv:2406.08929 (2024).
>
> The argument regarding why we approximate some distribution $\pi_{\text{approx}}$ is not convincing. The readers compare it with $\pi_\delta$ with early stopping.
>
> - Thank you for the comment. However, please note that in Theorem 3, we state that $\pi_{\text{approx}}$ is indeed $W_2$ close to $\pi_0$, which seems to have been overlooked.
>
> … In contrast, in the main text, how it is defined is never specified. When I check the proof (Lemma 6), I find that this distribution even depends on the transition kernel of $P_alg$. As a result, the kind of guarantee is very unusual. And the comparison with prior works under this metric is unfair.
>
> - We thank the reviewer for the insightful comment. We acknowledge that the guarantee we provide is subtle, so let us explain it in more detail here. Theorem 3 states that the law of the algorithm is KL close to $\pi_{\text{approx}}$, and that $\pi_{\text{approx}}$ is $W_2$ close to $\pi_0$. We note that since KL and $W_2$ are both upper bounds on the bounded Lipschitz (BL) metric, then our Theorem 3 implies that the law of the algorithm is close in BL distance to $\pi_0$, and the BL distance is a standard metric used to metrize weak convergence of probability measures (see [1]). However, we chose to state our guarantee in this way because it is slightly stronger.
>
> - Let us note that the original theoretical motivation for introducing the early stopped measure in Chen et al. (2023c) was a theoretical device. Indeed, it is generally impossible to obtain pure TV closeness to $\pi_0$, since this would entail learning the support exactly. However, diffusion models still enjoy convergence guarantees, provided that we weaken the guarantee slightly to allow the algorithm law to differ from $\pi_0$ both via vertical adjustments (TV/KL closeness) as well as horizontal adjustments ($W_2$ closeness). The early stopped measure $\pi_\delta$ in Chen et al. (2023c) is simply a theoretical construct to formalize this intuition. Since our constructed $\pi_{\text{approx}}$ can be made just as close to $\pi_0$ in $W_2$ as $\pi_\delta$, it plays the same role. From a practitioner’s perspective, either approach leads to the same end result, that the law of the algorithm is close to $\pi_0$ in the aforementioned sense. We argue that our approach is more flexible, as it allows us to capture the same notion of closeness without restricting ourselves to a particular choice of intermediate measure.
>
> [1] “Real analysis and probability” by R. M. Dudley
>
> The paper misses a closely related work [2] when introducing the analysis of DDIM.
>
> - We thank the reviewer for pointing this out. We will add this citation in our revised draft.

---

> ### Comment · Reviewer_wzAK · 2025-11-25
>
> Thanks for your response. I do miss the claim of the Wasserstein distance convergence, and now I think the current theorem makes sense.
>
> Additionally, I still hold some doubt about the $\gamma$ notation. After checking Conforti et al. [1], I find that the equation can be formulated in this form due to $\nabla \log \gamma^d =-x$, thus it's an equivalent transformation. What's the benefit of this transformation? *"This is helpful as the relative score satisfies some nice properties which permit more elegant bounds."* What do you mean by "some nice properties" if the equations are always equivalent?
>
> As for the terminology of DDPM, after skimming the provided tutorial, I do not find where they conflate the two concepts. Section 2.4 is only to connect DDPM with SDE, instead of claiming they are the same thing. Moreover, I believe a tutorial can use some inaccurate concepts to help audiences understand, while the writing of a conference paper should be more rigorous. Could you further list some papers, published in top-tier conferences, that use the same terminology?
>
> Best,
> Reviewer wzAK

---

### Official Review · Reviewer_NdkL · 2025-10-30

**Soundness:** 3
**Presentation:** 3
**Contribution:** 3
**Rating:** 6
**Confidence:** 2

**Summary:**

This paper presents an important theoretical result for SDE-based diffusion samplers (like DDPMs). Until now, the theoretical complexity for ODE-based samplers was well-studied, but the limit for SDE-based samplers was still O(d), which is linear to the data dimension d. This created a large gap from practice, where O(1) steps can make good samples. This paper uses a new analysis called Denoising Diffusion Randomized Midpoint Method (DDRaM) to prove for the first time that SDE-based DDPM samplers can achieve O(sqrt(d)) sublinear complexity.

**Strengths:**

1. The biggest contribution is providing the first sublinear O(sqrt(d)) proof for DDPM (SDE). This is a very important step forward in diffusion model theory. Many prior works (including Li & Jiao, ICLR 2025) achieved sublinear complexity for ODE (DDIM), but they could not solve the problem for DDPM because of its stochasticity. This paper successfully fills this important theoretical gap.

2. Besides the theory, the paper shows that the proposed DDRaM method works well in practice. In experiments on AFHQv2 (Figures 1, 2), DDRaM consistently shows better performance (lower FID, FD DINOv2) than standard samplers like Euler-Maruyama (EMD) or Exponential Euler (EED). This shows the proposed analysis is not just for theory but also has practical benefits.

**Weaknesses:**

My only one concern is about the decreasing importance of the DDPM sampler itself. In practice, many researchers are trying to develop samplers with very small NFE (like DDIM, DPM-Solvers) to make generation faster. Or, they train the model differently from the beginning (like Consistency Models or Rectified Flow). It is clear that proving sublinear complexity for DDPM was a very difficult problem, but I am a little unsure if solving this problem is as important as prior works on DDIM, which is more widely used in practice.

**Questions:**

In L1355 (Appendix C.2.1), you mentioned using numerical functions like `scipy.integrate.quad` and `scipy.optimize.root_scalar` for quadrature and root finding. Doesn't this add significant latency to the sampling process at each step?

---

> ### Author Response · Authors · 2025-11-19
> **Response to Reviewer NdkL**
>
> My only one concern is about the decreasing importance of the DDPM sampler itself. In practice, many researchers are trying to develop samplers with very small NFE (like DDIM, DPM-Solvers) to make generation faster. Or, they train the model differently from the beginning (like Consistency Models or Rectified Flow). It is clear that proving sublinear complexity for DDPM was a very difficult problem, but I am a little unsure if solving this problem is as important as prior works on DDIM, which is more widely used in practice.
>
> - The reviewer makes a valid point, but we argue that DDPM samplers are still preferred in some applications due to their higher empirical stability for small values of $\varepsilon$. See e.g. [1].
>
> - We use the terminology DDPM, but in actuality we analyze a prototypical SDE used for stochastic sampling. This naturally extends to other diffusion processes used in practice (VP, VE, EDM, etc.) as we show in our experiments. It will also extend to stochastic sampling with rectified flow, flow matching, or stochastic interpolants due to the need to solve an ODE or SDE for sampling in those frameworks as well (and because of the many structural similarities between flow-based and diffusion models). In short, we are not tied to the DDPM framework or sampler at all.
> - Beyond sampling from diffusion models, we emphasize that the randomized midpoint method is a general SDE solver with excellent accuracy and computation efficiency. Numerically solving SDEs is still a crucial component of many cutting-edge applications in fine-tuning [2], sampling [3], and molecular generation [4, 5], and we are interested in applying the randomized midpoint method in these settings and others in future work.
>
> [1] T. Deveney, et al. "Closing the ODE–SDE gap in score-based diffusion models through the Fokker–Planck equation." Philosophical Transactions A 383.2298 (2025): 20240503.
>
> [2] C. Domingo-Enrich, M. Drozdzal, B. Karrer, and R. T. Q. Chen, “Adjoint Matching: Fine-tuning Flow and Diffusion Generative Models with Memoryless Stochastic Optimal Control,” Jan. 07, 2025, arXiv: arXiv:2409.08861. doi: 10.48550/arXiv.2409.08861.
>
> [3] A. Havens et al., “Adjoint Sampling: Highly Scalable Diffusion Samplers via Adjoint Matching,” May 28, 2025, arXiv: arXiv:2504.11713. doi: 10.48550/arXiv.2504.11713.
>
> [4] T. Geffner et al., “Proteina: Scaling Flow-based Protein Structure Generative Models,” Mar. 02, 2025, arXiv: arXiv:2503.00710. doi: 10.48550/arXiv.2503.00710.
>
> [5] T. Geffner et al., “La-Proteina: Atomistic Protein Generation via Partially Latent Flow Matching,” Jul. 13, 2025, arXiv: arXiv:2507.09466. doi: 10.48550/arXiv.2507.09466.
>
> Questions:
> In L1355 (Appendix C.2.1), you mentioned using numerical functions like scipy.integrate.quad and scipy.optimize.root_scalar for quadrature and root finding. Doesn't this add significant latency to the sampling process at each step?
>
> - The quadrature and root finding are to determine the scalar coefficients with which the drift and noise are weighted. However, each step already involves two evaluations of the score function, which is a forward pass through a large neural network in applications. Although we have not rigorously benchmarked the wall-clock latency, we expect these numerical functions to have a negligible effect on runtime in theory and in practice, especially because of the nature of the smooth functions considered in this work.

---

### Official Review · Reviewer_91Te · 2025-11-01

**Soundness:** 2
**Presentation:** 3
**Contribution:** 1
**Rating:** 4
**Confidence:** 3

**Summary:**

This paper analyzes the denoising diffusion randomized midpoint method (DDRaM) — an SDE-based integrator for denoising diffusion probabilistic models (DDPMs), inspired by log-concave sampling (Shen & Lee, 2019). Using the "shifted composition rule" framework, it shows DDRaM needs sublinear score evaluations for convergence (the first sublinear complexity bound for pure DDPM sampling. Experimental validation confirms DDRaM works well with pre-trained image synthesis models.

**Strengths:**

- As far as I know, this is indeed the first $O(\sqrt{d})$ order error bound for a stochastic sampler in the diffusion model area.
- The paper is clearly written, and the idea is easy to follow.

**Weaknesses:**

- In the paper of Shen & Lee (2019), their analysis requires the target distribution to be a log-concave one. I am not sure if the Assumptions 1,2,3 in this paper can lead to the conclusion that the marginal distribution will be log-concave. Or could you explain how you could surpass this condition?
- The experiments validate the usage of the DDRaM method, but it does not involve popular stochastic samplers like DDPM itself, EDM-stochastic, PNDM-Stochastic, and DPM-Solver-Stochastic for comparison. Actually, I suppose these high-order stochastic samplers may also possess the property of requiring only sublinear score evaluations to ensure convergence. I encourage the authors to analyze these high-order samplers in practical use.

**Questions:**

Please refer to Weaknesses.

---

> ### Author Response · Authors · 2025-11-19
> **Response to Reviewer 91Te**
>
> In the paper of Shen & Lee (2019), their analysis requires the target distribution to be a log-concave one. I am not sure if the Assumptions 1,2,3 in this paper can lead to the conclusion that the marginal distribution will be log-concave. Or could you explain how you could surpass this condition?
>
> - Assumptions 1, 2, 3 do not require log-concavity or LSI for the target distribution, rather they only require time-varying Lipschitzness of the score. As shown in the original papers of Chen et al. (2023c) and Lee et al. (2023), guarantees for diffusion models do not require such assumptions because they rely on the fast mixing of the Ornstein–Uhlenbeck process. Intuitively, the effect of running the diffusion forward and then backward is similar to annealing and overcomes issues with multimodality. In comparison, the analysis of Shen and Lee (2019) is for a local Markov chain, which does suffer from slow mixing.
>
> The experiments validate the usage of the DDRaM method, but it does not involve popular stochastic samplers like DDPM itself, EDM-stochastic, PNDM-Stochastic, and DPM-Solver-Stochastic for comparison.
>
> - We thank the reviewer for pointing us to relevant samplers for comparison. We plan on adding a more detailed comparison to various stochastic sampling algorithms in the future, hopefully by the final revision deadline of December 3rd.
>
> Actually, I suppose these high-order stochastic samplers may also possess the property of requiring only sublinear score evaluations to ensure convergence. I encourage the authors to analyze these high-order samplers in practical use.
>
> - We thank the reviewer for the interesting suggestion. Higher-order solvers can indeed improve the accuracy, but they typically require higher-order derivative assumptions or incur worse dimension dependence for their analysis. See [1] and [2]. We emphasize that in the log-concave sampling literature, randomized midpoint discretization attains state-of-the-art bounds with regards to dimension dependence without further smoothness assumptions. Moreover, for discretizations of the overdamped Langevin diffusion, randomized midpoint is the only one to achieve iteration complexity sublinear in d. Therefore, at present, it seems out of reach (or is potentially impossible) to reach such results with higher-order solvers.
>
> [1] “Stochastic Runge–Kutta methods: provable acceleration of diffusion models” by Y. Wu, Y. Chen, and Y. Wei.
>
> [2] “Faster diffusion models via higher-order approximation” by G. Li, Y. Zhou, Y. Wei, and Y. Chen.

---

### Author Response · Authors · 2025-12-04
**Summary for the Area Chair**

## Overview of contributions
We would like to reiterate the significance and novelty of the main contributions of our work: we give the first proof that SDE-based diffusion models can sample in a number of iterations sublinear in the dimension. Previously, only ODE-based diffusion models were known to achieve sublinear complexity. To prove our result, we introduce a powerful new mathematical tool to the diffusion model theory literature, namely “shifted composition,” a method originating in the differential privacy and log-concave sampling communities. Additionally, we empirically demonstrate that the discretization scheme that we study can be implemented with minimal overhead and outperforms schemes widely used in practice like the Heun integrator.

## Overview of discussion
For the benefit of the metareviewer, we summarize the key points in our response.

- Although most reviewers were positive about the results of the paper in general, there were some misunderstandings around the theoretical guarantees. First, (1) Reviewer 91Te was concerned about the lack of a log-concavity assumption in our work. Reviewer wzAK raised additional points about (2) the reverse SDE formulation and (3) the nature of the guarantees (the role of the measure $\pi_{\operatorname{approx}}$). However, we clarify in our responses that these questions stem from misunderstandings of our results. In particular, we show that (1) and (2) are not anomalous, and are entirely analogous to prior work. (3) stems from a misunderstanding of our guarantees; the role of $\pi_{\operatorname{approx}}$ is precisely to show that our measure is $W_2$-close to $\operatorname{TV}$ close, which is analogous to the role of $\pi_\delta$ in other guarantees which utilize early stopping.
- Some reviewers requested additional comparisons in the experiments. While our existing experiments already show a marked improvement over the Euler and Heun integrators in the ODE setting, reviewers were interested in comparisons against other higher-order integrators, and with higher NFE budgets (particularly in the SDE setting). We have included these experiments in a revision (Appendix C.3, Stochastic sampling). In addition, we have added clarifications and additional experimental details as requested by the reviewers. In the process of the review, we have also found and corrected implementation bugs affecting Figures 1 and 2 (though our high-level conclusions remain the same).
- Finally, some high-level questions were raised concerning the importance of "DDPM"-type samplers. Here, we again emphasize the practical benefits of SDE-type samplers due to their stability at smaller noise levels, and also remark that our method can be applied to other SDE-based schemes. Reviewers were also concerned whether our algorithm is truly "simple" to implement. We show that it is comparable to the widely-used Heun scheme, which also uses two evaluations per step, and much more implementation-friendly when compared to predictor-corrector-type algorithms.

Please see our responses for more detailed comments.

---

### Meta-Review · Area_Chair_WjSN · 2026-01-07

**Summary:**

The submission proves that SDE samplers can attain $\tilde\Theta(\sqrt d)$ iterations of sampling just as ODE ones for the first time (which is previously believed to be $O(d)$). Which shows that stochasticity is not the bottleneck limiting DDPM sampling. The submission design a sampler with RMD and prove the sublinear iterations for the sampler. Experimental results show that samplers with RMD outperforms the ordinary SDE samplers for some degree. Reviewers' main concerns are about the assumptions, experiments and sampling schedule. After a productive discussion, the author has adequately dealt with reviewers' concerns.

**Reviewer Concerns:**

Addressed Concerns:

Reviewer 91Te:
  1. About the lack of a log-concavity assumption: Assumptions 1, 2, 3 do not require log-concavity or LSI for the target distribution.
  2. About results on more samplers: Newer results are updated in Appendix.

Reviewer NdkL:
  1. About the importance of SDE sampler: The SDE sampler remains some advantages. The submission mainly contributes to diffusion theory, and have possible extension to other solvers, flow matching and numerical computing.
  2. Latency introduced by numerical solver: The latency is negligible.

Reviewer wzAK:
  1. Question on reverse SDE form: The equation provided in the submission is equal to the usually used form.
  2. Inaccuracy in keyword "DDPM": The DDPM is used for referring to score-based SDE sampling for readability, and will be footnoted in revision.
  3. Mixture usage of $\pi_{approx}$ and $\pi_0$ in the theory:  $\pi_{approx}$ is $W_2$ closed to $\pi_0$ and the result in theory is appropriate.
  4. The distribution $\pi_{approx}$ depends on transition kernel $P_{alg}$, the comparison with prior may be unfair: The result is presented in a subtle but stronger way, which is appropriate. Also $\pi_{approx}$ can be made just as close to $\pi_0$ as $\pi_{\delta}$ in $W_2$, therefore abandoning the $\pi_{\delta}$ term offers extra flexibility.
  5. Missing related work: Will be added in revision.

Reviewer qw3a:
  1. Choice of decaying step size: Use empirical schedule step size in the comparison experiments.
  2. 2 NFE per step: This is usually in P-C sampler, and all comparison is made in NFE.

Remaining Concerns:

Reviewer 91Te:
  1. Analysis of high-order SDE sampler: Currently not possible.

Reviewer wzAK:
  1. Nice properties of the identical form of SDE.
  2. Questions about DDPM terminology.

**Reviewer Scores:**

Reviewer 91Te: rank 4, remaining inactive in rebuttal.

Reviewer NdkL: rank 6, remaining inactive in rebuttal, concerns are fully addressed.

Reviewer wzAK: rank 4, think that the theorem makes sense, could raise the rank for concerns being addressed.

Reviewer qw3a: rank 4, would be like to remain the rank.

The theoretical results in proving sublinear iterations of SDE sampler is reasonable. The experimental results in C.3 is not convincing enough in the performance of the proposed scheduler.

---

### Decision · Program_Chairs · 2026-01-26

Reject